# Faster Diffusion: Rethinking the Role of the Encoder for Diffusion Model Inference

**Senmao Li**[1*]**, Taihang Hu**[1*]**, Joost van de Weijer**[2]**, Fahad Shahbaz Khan**[3,4]**, Tao Liu**[1]
**Linxuan Li**[1]**, Shiqi Yang**[5]**, Yaxing Wang**[1†]  **Ming-Ming Cheng**[1]**,  Jian Yang**[1]

[1]VCIP, CS, Nankai University, [2]Computer Vision Center, Universitat Autònoma de Barcelona
[3]Mohamed bin Zayed University of AI, [4]Linkoping University, [5]Independent Researcher, Tokyo

{senmaonk, hutaihang00, ltolcy0, linxuanli520, shiqi.yang147.jp}@gmail.com
joost@cvc.uab.es,  fahad.khan@liu.se, {yaxing,cmm,csjyang}@nankai.edu.cn
https://sen-mao.github.io/FasterDiffusion

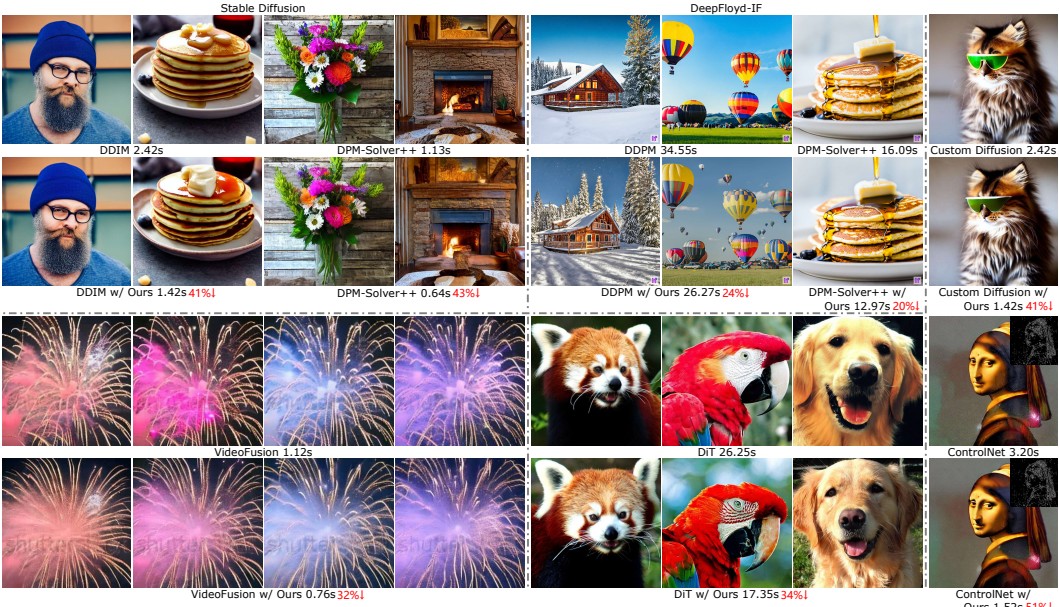

Figure 1: Results of our method for a diverse set of generation tasks. We significantly increase the image generation speed (second/image).

## Abstract

One of the main drawback of diffusion models is the slow inference time for image generation. Among the most successful approaches to addressing this problem are distillation methods. However, these methods require considerable computational resources. In this paper, we take another approach to diffusion model acceleration. We conduct a comprehensive study of the UNet encoder and empirically analyze the encoder features. This provides insights regarding their changes during the inference process. In particular, we find that encoder features change minimally, whereas the decoder features exhibit substantial variations across different time-steps. This insight motivates us to omit encoder computation at certain adjacent time-steps and reuse encoder features of previous time-steps as input to the decoder in multiple time-steps. Importantly, this allows us to perform decoder computation in parallel, further accelerating the denoising process. Additionally, we introduce a prior noise injection method to improve the texture details in the generated image. Besides the standard text-to-image task, we also validate our approach on

---

[*]Equal contribution. Author ordering determined by coin flip over a Google Hangout.
[†]The corresponding author.

38th Conference on Neural Information Processing Systems (NeurIPS 2024).

other tasks: text-to-video, personalized generation and reference-guided generation. Without utilizing any knowledge distillation technique, our approach accelerates both the Stable Diffusion (SD) and DeepFloyd-IF model sampling by 41% and 24% respectively, and DiT model sampling by 34%, while maintaining high-quality generation performance.

# 1 Introduction

One of the popular paradigms in image generation, Diffusion Models (DMs) [1, 2, 3] have recently achieved significant breakthroughs in various domains, including text-to-video generation [4, 5, 6], personalized image generation [7, 8, 9] and reference-guided image generation [10, 11, 12]. While diffusion models produce images of exceptional visual quality, their primary drawback lies in the prolonged inference time. The original diffusion model had an inference time several orders of magnitude slower than, for instance, GANs. One of the challenges hindering the acceleration of diffusion models is their inherent sequential denoising process, which limits the possibilities of effective parallelization.

To improve inference time speed of diffusion models, several methods have been developed, that can roughly be divided in two sets of approaches. Firstly, involving step reduction, the aim is to reduce the number of sampling steps within diffusion model inference, such as DDIM [13] and DPM-Solver [14], which have significantly reduced the number of sampling steps. Secondly, in contrast, knowledge distillation progressively distills a slow (many-step) teacher model into a faster (few-step) student model [15, 16]. Some recent works [17, 18, 19] excel at generating high-fidelity images in a few-step sampling scenario but face challenges in maintaining quality and diversity in one-step sampling. The main drawback of the distillation methods is that they require retraining to perform the distillation into faster diffusion models.

Orthogonal to these methods, we take a closer look at the sequential nature of the denoising process. We focus on the characteristics of the encoder in pretrained diffusion models (e.g., the SD and the DiT [21][2]) Interestingly, based on our analysis presented in Sec 3.2, we discover that encoder features change minimally (Fig. 3a) and have a high degree of similarity (Fig. 2 (top)), whereas the decoder features exhibit substantial variations across different time-steps (Fig. 3a and Fig. 2 (bottom)). This insight is relevant because it allows us to circumvent the computation of

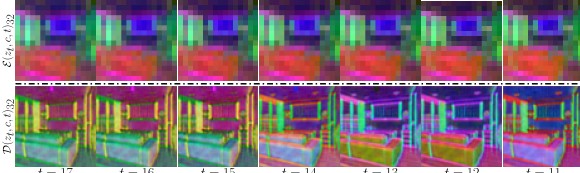

Figure 2: Visualising the hierarchical features [1]. We applied PCA to the hierarchical features following PnP [20] and used the top three leading components as an RGB image for visualization. The encoder features change minimally and have similarities at many time-steps (top), while the decoder features exhibit substantial variations across different time-steps (bottom).

the encoder during multiple time-steps. As a consequence, the decoder computations which are based on the same encoder input can be performed in parallel. Instead, we reuse the computed encoder features computed at one time-step (since these change minimally) as input to adjacent decoders during the following time-steps. More recently, both DeepCache[22] and CacheMe[23] leverage feature similarity to achieve acceleration. However, they rely on sequential denoising, as well as CacheMe requires fine-tuning. Unlike these methods, our approach supports parallel processing, which leads to faster inference (Tab. 2).

We show that the proposed propagation scheme accelerates the SD sampling by 24% , DeepFolyd-IF sampling by 18%, and DiT sampling by 27%. Furthermore, since the same encoder features (from previous time-steps) can be used as the input to the decoder of multiple later time-steps, this makes it possible to conduct multiple time-steps decoding concurrently. This parallel procession accelerates SD sampling by 41%, DeepFloyd-IF sampling by 24%, and DiT sampling by 34%. Furthermore, to alleviate the deterioration of the generated quality, we introduce a prior noise injection strategy to preserve the texture details in the generated images. With these contributions, our proposed method achieves improved sampling efficiency while maintaining high image generation quality.

---

[1]See the visualisation of the hierarchical features for all blocks in Appendix A

[2]We define the first several transformer blocks of DiT as the encoder, and the remaining ones of the transform blocks as the decoder.

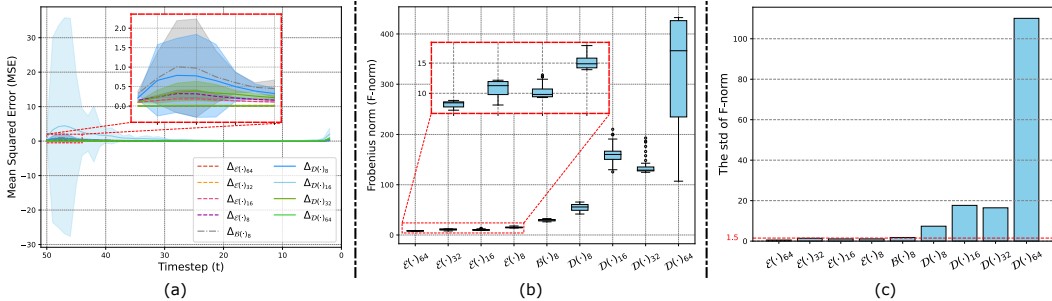

Figure 3: **Analyzing the UNet in Diffusion Model.** (a) Feature evolving across adjacent time-steps is measured by MSE. (b) We extract the hierarchical features output of different layers of the UNet at each time-step, average them along the channel dimension to obtain two-dimensional hierarchical features, and then calculate their *Frobenius norm.* (c) The hierarchical features of the UNet encoder show a lower standard deviation, while those of the decoder exhibit a higher standard deviation.

Importantly, our method can be combined with several approaches existing to speed up DMs. The main advantage of our method with respect to distillation-based approaches, is that our method can be applied at inference time, and does not require retraining a new faster distillation model; a process that is computationally very demanding and infeasible for actors with limited computational budget. Finally, we evaluate the effectiveness of our approach across a wide range of conditional diffusion-based tasks, including text-to-video generation (e.g., Text2Video-zero [4] and VideoFusion [5]), personalized image generation (e.g., Dreambooth [7]) and reference-guided image generation (e.g., ControlNet [10]).

To summarize, we make the following contributions:

- We conduct a thorough empirical study of the features of the UNet in the diffusion model showing that encoder features vary minimally (whereas decoder feature vary significantly).

- We propose a parallel strategy for diffusion model sampling at adjacent time-steps that significantly accelerates the denoising process. Importantly, our method does not require any training or fine-tuning technique.

- Furthermore, we also present a prior noise injection method to improve the image quality (mainly improving the quality of high-frequency textures).

- Our method can be combined with existing methods (like DDIM, and DPM-solver) to further accelerate diffusion model inference time.

## 2   Related Work

**Denoising diffusion model.**   Recently, Text-to-image diffusion models [1, 24, 25, 26] have made significant advancements. Notably, Stable Diffusion and DeepFloyd-IF stand out as two of the most successful diffusion models available within the current open-source community. These models, building upon the UNet architecture, are versatile and can be applied to a wide range of tasks, including image editing [27, 28], super-resolution [29, 30], segmentation [31, 32], and object detection [33, 34]. Given the strong scalability of transformer networks, DiT [21] investigates the transformer backbone for diffusion models.

**Diffusion model acceleration.**   Diffusion models use iterative denoising with UNet for image generation, which is time-consuming. There are plenty of works trying to address this issue. One strategy involves employing efficient diffusion model solvers, such as DDIM [13] and DPM-Solver [14], which have demonstrated notable reductions in sampling steps. Additionally, ToMe [35] exploits token redundancy to minimize the computations necessary for attention operations [36]. Conversely, knowledge distillation methods, exemplified by techniques like progressive simplification by student models [15, 16], aim to streamline existing models. Some recent studies combine model compression with distillation to achieve faster sampling [37, 38]. Orthogonal to these approaches, we introduce a novel method for enhancing sampling efficiency in DMs inference. We show that our method can be combined with several existing speed-up methods for further acceleration.

DeepCache[22] and CacheMe[23] are two recent works that leverage feature similarity to achieve acceleration. DeepCache [22] adopts a strategy of rudely reusing features cached from the previous step, requiring iterative denoising. Furthermore, CacheMe [23] requires additional fine-tuning for

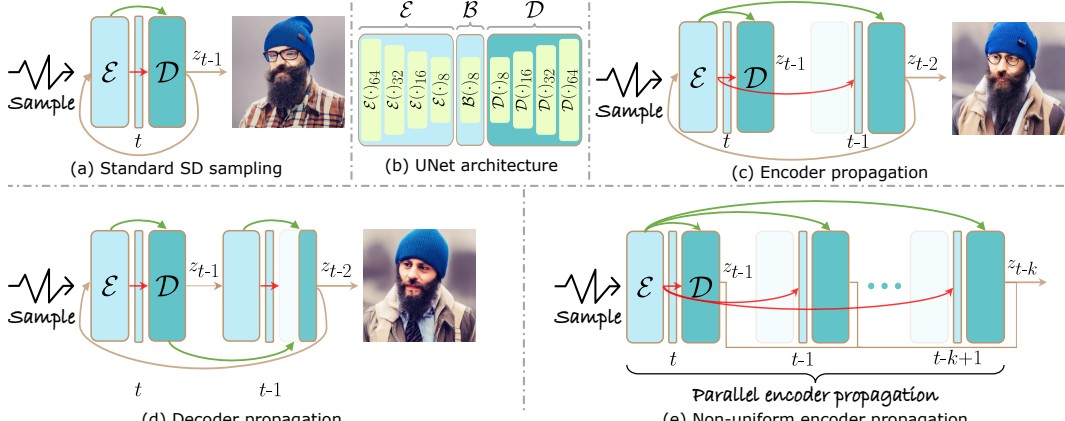

Figure 4: (a) Standard SD sampling. (b) UNet architecture. (c) Encoder propagation. We omit the encoder at certain adjacent time-steps and reuse in parallel the encoder features in the previous time-steps for the decoder. Applying encoder propagation for uniform strategy every two iterations. Note, at time-step $t$-1, predicting noise does not require $z_{t-1}$ (i.e., Eq. 1: $z_{t-2} = \sqrt{\frac{\alpha_{t-2}}{\alpha_{t-1}}} z_{t-1} + \sqrt{\alpha_{t-2}} \left( \sqrt{\frac{1}{\alpha_{t-2}} - 1} - \sqrt{\frac{1}{\alpha_{t-1}} - 1} \right) \cdot \epsilon_\theta(\cancel{z_{t-1}}, t-1, c)$). (d) Decoder propagation. The generated images often fail to cover some specific objects in the text prompt. For example, given one prompt case "A man with a beard wearing glasses and a beanie", this method fails to generate the **glasses** subject. See Appendix F for quantitative evaluation. (e) Applying encoder propagation for non-uniform strategy. By benefiting from our propagation scheme, we are able to perform the decoder in parallel at certain adjacent time-steps.

better performance. In contrast, our approach enables parallel processing, leading to considerably faster inference.

## 3 Method

We first briefly revisit the architecture of the Stable Diffusion (SD) (Sec. 3.1), and then conduct a comprehensive analysis for the hierarchical features of the UNet (Sec. 3.2). Our analysis shows that it is possible to parallelize the diffusion model denoising process partially. Thus, we introduce a novel method to accelerate the diffusion sampling while still largely maintaining the generation quality and fidelity (Sec. 3.3).

### 3.1 Latent Diffusion Model

In the diffusion inference stage, the denoising network $\epsilon_\theta$ takes as input a text embedding $c$, a latent code $z_t$ and a time embedding, predicts noise, resulting in a latent $z_{t-1}$ using the DDIM scheduler [13]:

$$z_{t-1} = \sqrt{\frac{\alpha_{t-1}}{\alpha_t}} z_t + \sqrt{\alpha_{t-1}} \left( \sqrt{\frac{1}{\alpha_{t-1}} - 1} - \sqrt{\frac{1}{\alpha_t} - 1} \right) \cdot \epsilon_\theta(z_t, t, c), \qquad (1)$$

where $\alpha_t$ is a predefined scalar function at time-step $t$ ($t = T, ..., 1$). The typical denoising network uses a UNet-based architecture. It consists of an encoder $\mathcal{E}$, a bottleneck $\mathcal{B}$, and a decoder $\mathcal{D}$, respectively (Fig. 4b). The hierarchical features extracted from the encoder $\mathcal{E}$ are injected into the decoder $\mathcal{D}$ by a skip connection (Fig. 4a). For convenience of description, we divide the UNet network into specific blocks: $\mathcal{E} = \{\mathcal{E}(\cdot)_s\}$, $\mathcal{B} = \{\mathcal{B}(\cdot)_8\}$, and $\mathcal{D} = \{\mathcal{D}(\cdot)_s\}$, where $s \in \{8, 16, 32, 64\}$ (see Fig. 4b). Both $\mathcal{E}(\cdot)_s$ [3] and $\mathcal{D}(\cdot)_s$ represent the block layers with input resolution $s$ in both encoder and decoder, respectively.

Diffusion Transformer (DiT) [21] is a novel architecture for diffusion models. It replaces the UNet backbone with a transformer, which consists of 28 blocks. Based on our observations, we define the first 18 blocks as the encoder, and the remaining 10 blocks as the decoder (see Appendix A.3).

### 3.2 Analyzing the UNet in Diffusion Model

In this section, we take the UNet-based diffusion model as example to analyze the properties of the pretrained diffsuion model. We delve into the UNet which consists of the encoder $\mathcal{E}$, the bottleneck $\mathcal{B}$,

---

[3]Once we replace the $\cdot$ with specific inputs in $\mathcal{E}(\cdot)_s$, we define that it represents the feature of $\mathcal{E}(\cdot)_s$

and the decoder $\mathcal{D}$, for a deeper understanding of the different parts of the UNet. Note the following observed properties also exist in DiT (see Appendix A.3).

**Feature evolution across time-steps.** We experimentally observe that the encoder features exhibit a subtle variation at adjacent time-steps, whereas the decoder features exhibit substantial variations across different time-steps (see Fig. 3a and Fig. 2). Specifically, given a pretrained diffusion model, we iteratively produce a latent code $z_t$ (see Eq. 1), and the corresponding hierarchical features: $\{\mathcal{E}(z_t, c, t)_s\}$, $\{\mathcal{B}(z_t, c, t)_8\}$, and $\{\mathcal{D}(z_t, c, t)_s\}$ ($s \in \{8, 16, 32, 64\}$) [4], as shown in Fig. 4b. Here, we analyze how the hierarchical features change at adjacent time-steps. To achieve this goal, we quantify the variation of the hierarchical features as follows:

$$\Delta_{\mathcal{E}(\cdot)_s} = \frac{1}{d \times s^2} \|\mathcal{E}(z_t, c, t)_s - \mathcal{E}(z_{t-1}, c, t-1)_s\|_2^2, \tag{2}$$

where $d$ represents the number of channels in $\mathcal{E}(z_t, c, t)_s$. Similarly, we also compute $\Delta_{\mathcal{B}(\cdot)_8}$ and $\Delta_{\mathcal{D}(\cdot)_s}$.

As illustrated in Fig. 3a, for both the encoder $\mathcal{E}$ and the decoder $\mathcal{D}$, the curves exhibit a similar trend: in the wake of an initial increase, the variation reaches a plateau and then decreases, followed by a continuing growth towards the end. However, the extent of change in $\Delta_{\mathcal{E}(\cdot)_s}$ and $\Delta_{\mathcal{D}(\cdot)_s}$ is quantitatively markedly different. For example, the maximum value and variance of the $\Delta_{\mathcal{E}(\cdot)_s}$ are less than 0.4 and 0.05, respectively (Fig. 3a (zoom-in area)), while the corresponding values of the $\Delta_{\mathcal{D}(\cdot)_s}$ are about 5 and 30, respectively (Fig. 3a). Furthermore, we find that $\Delta_{\mathcal{D}(\cdot)_{64}}$, the change of the last layer of the decoder, is close to zero. This is due to the output of the denoising network being similar at adjacent time-steps [39]. In conclusion, the overall feature change $\Delta_{\mathcal{E}(\cdot)_s}$ is smaller than $\Delta_{\mathcal{D}(\cdot)_s}$ throughout the inference phase.

**Feature evolution across layers.** We experimentally observe that the feature characteristics are significantly different between the encoder and the decoder across all time-steps. For the encoder $\mathcal{E}$ the intensity of the change is slight, whereas it is very drastic for the decoder $\mathcal{D}$. Specifically we calculate the *Frobenius norm* for hierarchical features $\mathcal{E}(z_t, c, t)_s$ across all time-steps, dubbed as $\mathcal{F}_{\mathcal{E}(\cdot)_s} = \{\mathcal{F}_{\mathcal{E}(z_T, c, T)_s}, ..., \mathcal{F}_{\mathcal{E}(z_1, c, 1)_s}\}$. Similarly, we compute $\mathcal{F}_{\mathcal{B}(\cdot)_8}$ and $\mathcal{F}_{\mathcal{D}(\cdot)_s}$, respectively.

Fig. 3b shows the feature evolution across layers with a boxplot [5]. Specifically, for $\{\mathcal{F}_{\mathcal{E}(\cdot)_s}\}$ and $\{\mathcal{F}_{\mathcal{B}(\cdot)_8}\}$, the box is relatively compact, with a narrow range between their first-quartile and third-quartile values. For example, the maximum box height ($\mathcal{F}_{\mathcal{E}}(.)_{32}$) of these features is less than 5 (see Fig. 3b (zoom-in area)). This indicates that the features from both the encoder $\mathcal{E}$ and the bottleneck $\mathcal{B}$ slightly change. In contrast, the box heights corresponding to $\{\mathcal{D}(\cdot)_s\}$ are relatively large. For example, for the $\mathcal{D}(.)_{64}$ the box height is over 150 between the first quartile and third quartile values (see Fig. 3b). Furthermore, we also provide a standard deviation (Fig. 3c), which exhibits similar phenomena to Fig. 3b. These results show that the encoder features have relatively small discrepancy and a high degree of similarity across all layers. However, the decoder features evolve drastically.

**Could we omit the Encoder at certain time-steps?** As indicated by the previous experimental analysis, we observe that, during the denoising process, the decoder features change drastically, whereas the encoder $\mathcal{E}$ features change minimally, and have a high degree of similarities at certain adjacent time-steps. Therefore, as shown in Fig. 4c, We propose to omit the encoder at certain time-steps and use the same encoder features for several decoder steps. This allows us to compute these multiple decoder steps in parallel.

Specifically, we delete the encoder at time-step $t - 1$ ($t - 1 < T$), and the corresponding decoder (including the skip connections) takes as input the hierarchical outputs of the encoder $\mathcal{E}$ from the previous time-step $t$, instead of the ones from the current time-step $t-1$ like the standard SD sampling (for more detail, see Sec. 3.3).

When omitting the encoder at a certain time-step, we are able to generate similar images (Fig. 4c) like standard SD sampling (Fig. 4a, Tab. 1 (the first and second rows) and additional results in Appendix F). Alternatively, if we use a similar strategy for the decoder (i.e., *decoder propagation*), we find the

---

[4]The feature resolution is half of the previous one in the encoder and two times in the decoder. Note that the feature resolutions of $\mathcal{E}(.)_8$, $\mathcal{B}(.)_8$ and $\mathcal{D}(.)_{64}$ do not change in the SD model.

[5]Each boxplot contains the minimum (0th percentile), the maximum (100th percentile), the median (50th percentile), the first quartile (25th percentile) and the third quartile (75th percentile) values of the feature Frobenius norm (e.g., $\{\mathcal{F}_{\mathcal{E}(z_T, c, T)_s}, ..., \mathcal{F}_{\mathcal{E}(z_1, c, 1)_s}\}$).

generated images often fail to cover some specific objects in the text prompt (Fig. 4d). For example, when provided with prompt "A man with a beard wearing glasses and a beanie", the SD model fails to synthesize "glasses" when applying decoder propagation. This is due to the fact that the semantics are mainly contained in the features from the decoder rather than the encoder [40].

The *encoder propagation*, which uses encoder outputs from previous time-step as the input to the current decoder, could speed up the diffusion model sampling at inference time. In the following Sec. 3.3, we give further elaborate on encoder propagation.

## 3.3 Encoder propagation

Diffusion sampling, combining iterative denoising with transformers, is time-consuming. Therefore we propose a novel and practical diffusion sampling acceleration method. During the diffusion sampling process $t = \{T, ..., 1\}$, we refer to the time-steps where encoder propagation is deployed, as *non-key* time-steps denoted as $t^{non\text{-}key} = \left\{ t_0^{non\text{-}key}, ..., t_{N-1}^{non\text{-}key} \right\}$. The remaining time-steps are dubbed as $t^{key} = \left\{ t_0^{key}, t_1^{key}, ..., t_{T-1-N}^{key} \right\}$. In other words, we do not use the encoder at time-steps $t^{non\text{-}key}$, and leverage the hierarchical features of the encoder from the time-step $t^{key}$. Note we utilize the encoder $\mathcal{E}$ at the initial time-step ($t_0^{key} = T$). Thus, the diffusion inference time-steps could be reformulated as $\{t^{key}, t^{non\text{-}key}\}$, where $t^{key} \cup t^{non\text{-}key} = \{T, ..., 1\}$ and $t^{key} \cap t^{non\text{-}key} = \varnothing$. In the following, we introduce both *uniform encoder propagation* and *non-uniform encoder propagation* strategies.

As shown in Fig. 3a, The encoder feature change is larger in the initial inference phase compared to the later phases throughout the inference process. Therefore, we select more *key* time-steps in the initial inference phase, and less *key* time-steps in later phases. We experimentally define the *key* time-steps as $t^{key} = \{50, 49, 48, 47, 45, 40, 35, 25, 15\}$ for SD model with DDIM, and $t^{key} = \{100, 99, 98, ..., 92, 91, 90, 85, 80, ..., 25, 20, 15, 14, 13, ..., 2, 1\}$[6], $\{50, 49, ..., 2, 1\}$ and $\{75, 73, 70, 66, 61, 55, 48, 40, 31, 21, 10\}$ for three stage of DeepFloyd-IF (see detail *key* time-steps selection in Appendix F.2). The remaining time-steps are categorized as *non-key* time-steps. We define this strategy as non-uniform encoder propagation (see Fig. 4e). As shown in Fig. 4c, we also explore the *non-key* time-step selection with fix stride (e.g, 2), dubbed as *uniform encoder propagation*.

Note that our method does not reduce the number of sampling steps. During encoder propagation, the decoder computes for all time-steps, necessitating time embedding inputs for each time-step to maintain temporal coherence (see detail in Appendix D).

Tab. 5 reports the results of the ablation study, considering various combinations of *key* and *non-key* time-steps. These results indicate that the set of non-uniform *key* time-steps performs better in generating images.

**Parallel non-uniform encoder propagation.** When applying the non-uniform encoder propagation strategy, at time-step $t \in t^{non-key}$ the decoder inputs do not rely on the encoder outputs at time-step $t$ (see Fig. 4e). Instead, it relies on the encoder output at the previous nearest *key* time-step. This allows us to perform *parallel non-uniform encoder propagation* at these adjacent time-steps in $t^{non-key}$. We perform decoding in parallel from $t$ to $t - k + 1$ time-steps. This technique further improves the inference

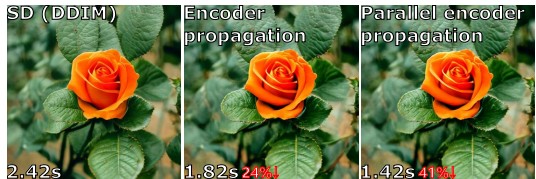

Figure 5: Comparing with SD (left), encoder propagation reduces the sampling time by $24\%$ (middle). Furthermore, parallel encoder propagation achieves a $41\%$ reduction in sampling time (right).

efficiency since the decoder forward in multiple time-steps could be conducted concurrently. We indicate this as *parallel-batch non-key* time-steps. As shown in Fig. 5 (right), this further reduces evaluation time by $41\%$ for the SD model.

**Prior noise injection.** Although the encoder propagation could improve the efficiency in the inference phase, we observe that it leads to a slight loss of texture information in the generated results (see Fig. 6 (left, middle)). Inspired by related works [41, 42], we propose a prior noise

---

[6]The ellipsis in $t^{key}$ denotes each time-step between the time-steps on either side of the ellipsis. For example, 80...25 means that every time-step between 80 and 25 is included.

injection strategy. It combines the initial latent code $z_T$ into the generative process at subsequent time-step (i.e., $z_t$), following $z_t = z_t + \alpha \cdot z_T$, if $t < \tau$, where $\alpha = 0.003$ is the scale parameter to control the impact of $z_T$. And we start to use this injection mechanism from $\tau = 25$ step. This strategic incorporation successfully improves the texture information. Importantly, it demands almost negligible extra computational resources. We observe that the loss of texture information occurs in all frequencies of the frequency domain (see Fig. 6 (right, red, and blue curves)). This approach ensures a close resemblance of generated results in the frequency domain to both SD and $z_T$ injection (see Fig. 6 (right, red and green curves)), with the generated images maintaining the desired fidelity (see Fig. 6 (left, bottom)).

## 4 Experiments

In our experiments, we assess the speed-up of our method compared to others for inference acceleration. We also explore combining our method with these approaches. We do not directly compare our method with distillation methods, which offer superior results but involve computationally expensive retraining.

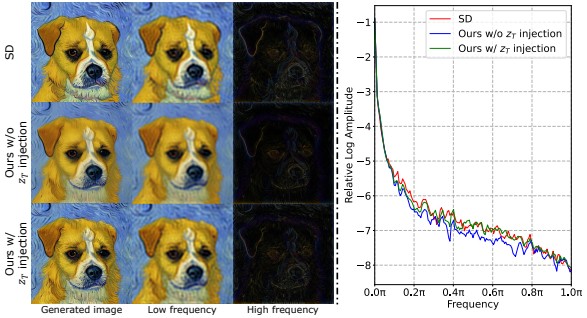

**Datasets and evaluation metrics.** We randomly select 10K prompts from the MS-COCO2017 validation dataset [43] and feed them into the text-to-image diffusion model to obtain 10K generated images. For the transformer architecture diffusion model, we randomly generate 50K images from 1000 ImageNet [44] class labels. For

Figure 6: (left) We keep the image content through $z_T$ injection, slightly compensating for the texture information loss caused by encoder propagation. (right) The amplitudes of the generated image through $z_T$ injection closely resemble those from SD.

other tasks, we use the same settings as baselines (e.g., Text2Video-zero [4], VideoFusion [5], Dreambooth [7] and ControlNet [10]). We use the Fréchet Inception Distance (FID) [45] metric to assess the visual quality of the generated images, and the Clipscore [46] to measure the consistency between image content and text prompt. Furthermore, we report the average values for both the computational workload (GFLOPs/image) and sampling time (s/image) to represent the resource demands for a single image. See more detailed implementation information on Appendix A.

### 4.1 Text-to-image Generation

We first evaluate the proposed encoder propagation method for the standard text-to-image generation task on both the latent space (i.e., SD) and pixel space (i.e., DeepFloyd-IF) diffusion models. As shown in Tab. 1, we significantly accelerate the diffusion sampling with negligible performance degradation. Specifically, our proposed method decreases the computational burden (GFLOPs) by a large margin (27%) and greatly reduces sampling time to 41% when compared to standard DDIM sampling in SD. Similarly, in DeepFloyd-IF, the reduction in both computational burden and time reaches 15% and 24%, respectively. Furthermore, our method can be combined with the latest sampling techniques like DPM-Solver [14], DPM-Solver++ [47], and ToMe [35]. Our method enhances sampling efficiency while preserving good model perfor-

Table 1: Quantitative evaluation[7] for both SD and DeepFloyd-IF diffusion models.

| DM | Sampling Method | T | FID↓ | Clip-score↑ | GFLOPs/ image↓ | s/image ↓ Unet of DM | s/image ↓ DM |
|---|---|---|---|---|---|---|---|
| Stable Diffusion | DDIM | 50 | 21.75 | 0.773 | 37050 | 2.23 | 2.42 |
| | DDIM w/ Ours | 50 | 21.08 | 0.783 | 27350 27%↓ | 1.21 45%↓ | 1.42 41%↓ |
| | DPM-Solver | 20 | 21.36 | 0.780 | 14821 | 0.90 | 1.14 |
| | DPM-Solver w/ Ours | 20 | 21.25 | 0.779 | 11743 21%↓ | 0.46 48%↓ | 0.64 43%↓ |
| | DPM-Solver++ | 20 | 20.51 | 0.782 | 14821 | 0.90 | 1.13 |
| | DPM-Solver++ w/ Ours | 20 | 20.76 | 0.781 | 11743 21%↓ | 0.46 48%↓ | 0.64 43%↓ |
| | DDIM + ToMe | 50 | 22.32 | 0.782 | 35123 | 2.07 | 2.26 |
| | DDIM + ToMe w/ Ours | 50 | 20.73 | 0.781 | 26053 26%↓ | 1.15 44%↓ | 1.33 41%↓ |
| DeepFloyd-IF | DDPM | 225 | 23.89 | 0.783 | 734825 | 33.91 | 34.55 |
| | DDPM w/ Ours | 225 | 23.73 | 0.782 | 626523 15%↓ | 25.61 25%↓ | 26.27 24%↓ |
| | DPM-Solver++ | 100 | 20.79 | 0.784 | 370525 | 15.19 | 16.09 |
| | DPM-Solver++ w/ Ours | 100 | 20.85 | 0.785 | 313381 15%↓ | 12.02 21%↓ | 12.97 20%↓ |

mance, with negligible variations of both FID and Clipscore values (Tab. 1 (the third to eighth rows)). Our method achieves good performance across different sampling steps (Fig. 7 and see Appendix D for quantitative results.). Importantly, these results show that our method is orthogonal and compatible

---
[7]We use the official implementation of Clipscore [46] to obtain around 0.75 but around 0.3. See Appendix C.

Table 2: Comparison with DeepCache and CacheMe. CacheMe is not open-source.

| Sampling Method | T | Parallel | FID ↓ | Clipscore ↑ | s/image |
|---|---|---|---|---|---|
| DDIM | 50 | ✗ | 21.75 | 0.773 | 2.42 |
| DDIM w/ DeepCache | 50 | ✗ | 21.53 | 0.770 | $1.05_{56\%\downarrow}$ |
| DDIM w/ CacheMe | 50 | ✗ | – | – | $1.30_{44\%\downarrow}$ |
| DDIM w/ Ours | 50 | ✓ | 21.62 | 0.775 | $0.56_{77\%\downarrow}$ |

Table 3: Quantitative evaluation for DiT.

| Sampling Method | T | Image Res. | FID ↓ | sFID ↓ | IS ↑ | Precision ↑ | Recall ↑ | s/image |
|---|---|---|---|---|---|---|---|---|
| DiT | 250 | 256 | 2.27 | 4.60 | 278.24 | 0.83 | 0.57 | 5.13 |
| DiT w/ Ours | 250 | 256 | **2.31** | **4.55** | 276.05 | **0.82** | **0.57** | $3.62_{29\%\downarrow}$ |
| DiT | 250 | 512 | 3.04 | 5.02 | 240.82 | 0.84 | 0.54 | 26.25 |
| DiT w/ Ours | 250 | 512 | **3.25** | **5.05** | 245.13 | **0.83** | **0.51** | $17.35_{34\%\downarrow}$ |

Table 4: Quantitative evaluation on text-to-video, personalized generation and reference-guided generation tasks. † and ‡ indicate "edges" and "scribble" conditions, respectively.

| Method | T | FID↓ | Clip-score↑ | GFLOPs/image↓ | s/image↓ Unet of SD | s/image↓ SD |
|---|---|---|---|---|---|---|
| Text2Video-zero | 50 | - | 0.732 | 39670 | 12.59/8 | 13.65/8 |
| Text2Video-zero w/ Ours | 50 | - | 0.731 | $30690_{22\%\downarrow}$ | $9.46/8_{25\%\downarrow}$ | $10.54/8_{23\%\downarrow}$ |
| VideoFusion | 50 | - | 0.700 | 224700 | 16.71/16 | 17.93/16 |
| VideoFusion w/ Ours | 50 | - | 0.700 | $148680_{33\%\downarrow}$ | $11.1/16_{34\%\downarrow}$ | $12.2/16_{32\%\downarrow}$ |
| ControlNet (†) | 50 | 13.78 | 0.769 | 49500 | 3.09 | 3.20 |
| ControlNet (†) w/ Ours | 50 | 14.65 | 0.767 | $31400_{37\%\downarrow}$ | $1.43_{54\%\downarrow}$ | $1.52_{51\%\downarrow}$ |
| ControlNet (‡) | 50 | 16.17 | 0.775 | 56850 | 3.85 | 3.95 |
| ControlNet (‡) w/ Ours | 50 | 16.42 | 0.775 | $35990_{37\%\downarrow}$ | $1.83_{53\%\downarrow}$ | $1.93_{51\%\downarrow}$ |
| Dreambooth | 50 | - | 0.640 | 37050 | 2.23 | 2.42 |
| Dreambooth w/ Ours | 50 | - | 0.660 | $27350_{27\%\downarrow}$ | $1.21_{45\%\downarrow}$ | $1.42_{41\%\downarrow}$ |
| CustomDiffusion | 50 | - | 0.640 | 37050 | 2.21 | 2.42 |
| CustomDiffusion w/ Ours | 50 | - | 0.650 | $27350_{27\%\downarrow}$ | $1.21_{45\%\downarrow}$ | $1.42_{41\%\downarrow}$ |

Table 5: Quantitative evaluation in various propagation strategies on MS-COCO 2017 10K subset. FTC=FID×Time/Clipscore.

| Propagation strategy | | FID ↓ | Clipscore ↑ | GFLOPs/image ↓ | s/image ↓ Unet of SD | s/image ↓ SD | FTC ↓ |
|---|---|---|---|---|---|---|---|
| SD | | 21.75 | 0.773 | 37050 | 2.23 | 2.42 | 68.1 |
| Uniform | I | $t^{key} = \{50, 48, 46, 44, 42, 40, 38, 36, 34, 32, 30, 28, 26, 24, 22, 20, 18, 16, 14, 12, 10, 8, 6, 4, 2\}$ | | | | | |
| | | 21.55 | 0.775 | $31011_{16\%\downarrow}$ | $1.62_{27\%\downarrow}$ | $1.81_{25\%\downarrow}$ | 50.3 |
| | II | $t^{key} = \{50, 44, 38, 32, 26, 20, 14, 8, 2\}$ | | | | | |
| | | 21.54 | 0.773 | $27350_{27\%\downarrow}$ | $1.26_{43\%\downarrow}$ | $1.46_{40\%\downarrow}$ | 40.7 |
| | III | $t^{key} = \{50, 38, 26, 14, 2\}$ | | | | | |
| | | 24.61 | 0.766 | $26370_{29\%\downarrow}$ | $1.12_{50\%\downarrow}$ | $1.36_{44\%\downarrow}$ | 43.7 |
| Non-uniform | I | $t^{key} = \{50, 40, 39, 38, 30, 25, 20, 15, 5\}$ | | | | | |
| | | 22.94 | 0.776 | $27350_{27\%\downarrow}$ | $1.26_{43\%\downarrow}$ | $1.42_{41\%\downarrow}$ | 41.9 |
| | II | $t^{key} = \{50, 30, 25, 20, 15, 14, 5, 4, 3\}$ | | | | | |
| | | 35.25 | 0.742 | $27350_{27\%\downarrow}$ | $1.25_{43\%\downarrow}$ | $1.42_{41\%\downarrow}$ | 67.4 |
| | III | $t^{key} = \{50, 41, 37, 35, 22, 21, 18, 14, 5\}$ | | | | | |
| | | 22.14 | 0.778 | $27350_{27\%\downarrow}$ | $1.22_{45\%\downarrow}$ | $1.42_{41\%\downarrow}$ | 40.4 |
| | IV (Ours) | $t^{key} = \{50, 49, 48, 47, 45, 40, 35, 25, 15\}$ | | | | | |
| | | 21.08 | 0.783 | $27350_{27\%\downarrow}$ | $1.21_{45\%\downarrow}$ | $1.42_{41\%\downarrow}$ | **38.2** |

with these acceleration techniques. As shown in Fig. 1, we visualize the generated images with different sampling techniques. Our method still generates high-quality results (see Appendix F for additional results).

Our method allows to use multi-GPU to generate one image. With multi-GPU parallel, our proposed method further accelerates the SD sampling by 77%, whereas DeepCache [22] and CacheMe [23] achieve speedups of 56% and 44%, respectively (See Tab. 2). These results indicate that we achieve superior acceleration compared to DeepCache [22] and CacheMe [23].

## 4.2 Diffusion Transformer

We also evaluate our approach on DiT. As reported in Tab. 3, we achieve accelerations of about 29% and 34% for DiT sampling with image resolution of 256 and 512, respectively, while preserving high-quality results (see Figs. 1 and 18).

## 4.3 Other tasks with text-guided diffusion model

Besides the standard text-to-image task, we also validate our proposed approach on other tasks: *text-to-video generation*, *personalized generation*, and *reference-guided image generation*.

**Text-to-video.** To evaluate our method, we combine it with both Text2Video-zero [4] and VideoFusion [5]. As reported in Tab. 4 (the second and fourth rows), when combined with our method, the two methods have a reduction of approximately 22% to 33% in both computational burden and generation time. These results indicate that we are able to enhance the efficiency of generative processes in the text-to-video task while preserving video fidelity at the same time (Fig. 1 (left, bottom)). As an example, when generating a video using the prompt "Fireworks bloom in the night sky", the VideoFusion model takes 17.92 seconds with 16 frames for the task (1.12s/frame), when combined with our method it only takes 12.27s (0.76s/frame) to generate a high-quality video (Fig. 1 (left, bottom)).

**Personalized image generation.** Dreambooth [7] and Custom Diffusion [8] are two approaches for customizing tasks by fine-tuning text-to-image diffusion models. As reported in Tab. 4 (the ninth to twelfth rows), our method, combining with the two customization approaches, accelerates image

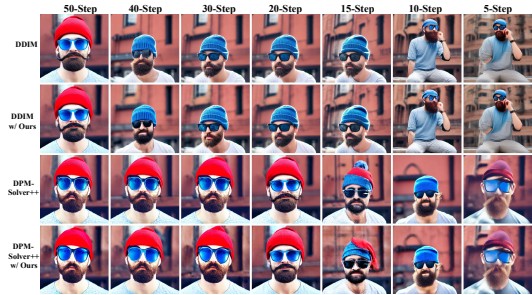

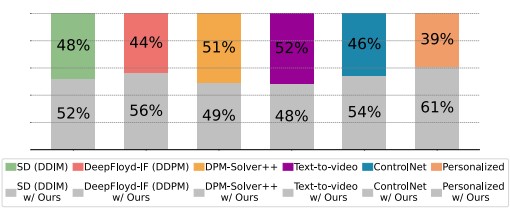

Figure 8: User study results.

Figure 7: Generated images at different time-steps.

generation and reduces computational demands. Visually, it maintains the ability to generate images with specific contextual relationships based on reference images. (Fig. 1 (right))

**Reference-guided image generation.** ControlNet [10] incorporates a trainable encoder, successfully generates a text-guided image, and preserves similar content with conditional information. Our approach can be applied concurrently to two encoders of ControNet. In this paper, we validate the proposed method with two conditional controls: *edge* and *scribble*. Tab. 4 (the fifth to eighth row) reports quantitative results. We observe that it leads to a significant decrease in both generation time and computational burden. Furthermore, Fig. 1 (middle, bottom) qualitatively shows that our method successfully preserves the given structure information and achieves similar results as ControlNet.

**User study.** We conducted a user study, as depicted in Fig. 8, and asked subjects to select results. We apply pairwise comparisons (forced choice) with 18 users (35 pairs of images or videos/user). The results demonstrate that our method performs equally well as the baseline methods.

### 4.4 Ablation study

We ablate the results with different selections of both uniform and non-uniform encoder propagation. Tab. 5 reports that the performance of the non-uniform setting outperforms the uniform one in terms of both FID and Clipscore (see Tab. 5 (the third and eighth rows)). Furthermore, we explore different configurations within the non-uniform strategy. The strategy, using the set of *key* time-steps we established, yields better results in the generation process (Tab. 5 (the eighth row)). We further present qualitative results stemming from the above choices. As shown in Fig. 9, given the same number of *key* time-steps, the appearance of nine-step non-uniform strategy $\mathbb{I}$, $\mathbb{II}$ and $\mathbb{III}$ settings do not align with the prompt "Fireflies dot the night sky". Although the generated image in the two-step setting exhibits a pleasing visual quality, its sampling efficiency is lower than our chosen setting (see Tab. 5 (the second and eighth rows)).

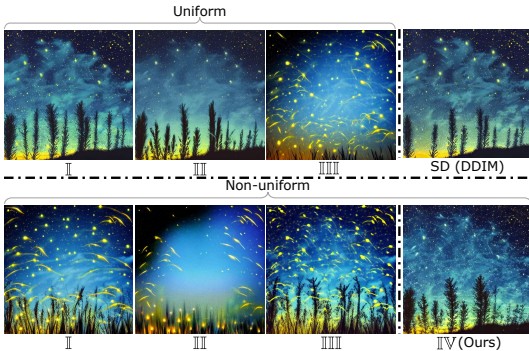

Figure 9: Generating image with uniform and non-uniform encoder propagation. The result of uniform strategy $\mathbb{II}$ yields smooth and loses textual compared with SD. Both uniform strategy $\mathbb{III}$ and non-uniform strategy $\mathbb{I}$, $\mathbb{II}$ and $\mathbb{III}$ generate images with unnatural saturation levels.

**Effectiveness of prior noise injection.** We evaluate the effectiveness of injecting initial $z_T$. As reported in Tab. 6, the differences in FID and Clipscores without $z_T$ (the third column), when compared to DDIM and Ours (the second and fourth columns), are approximately $0.01\%$, which can be considered negligible. While this

Table 6: Quantitative evaluation for prior noise injection.

| Sampling Method | SD (DDIM) | SD (DDIM) + Ours w/o $z_T$ injection | SD (DDIM) + Ours w/ $z_T$ injection |
|---|---|---|---|
| **FID** ↓ | 21.75 | 21.71 | 21.08 |
| **Clipscore** ↑ | 0.773 | 0.779 | 0.783 |

is not the case for the visual expression of the generated image, it is observed that the output contains complete semantic information with smoothing texture (refer to Fig. 6 (left, the second row)). Injecting the $z_T$ aids in maintaining fidelity in the generated results during encoding propagation (see Fig. 6 (left, the third row) and Fig. 6 (right, red and green curves)).

# 5  Conclusion

In this work, We explore the characteristics of the encoder and decoder in UNet of the text-to-image diffusion model and find that encoder feature variation is minimal for many time-steps, while the decoder plays a significant role across all time-steps. Building upon this finding, we propose encoder propagation for efficient diffusion sampling, reducing time on both the UNet-based and the transform-based diffusion models on a diverse set of generation tasks. We conduct extensive experiments and validate that our approach can achieve improved sampling efficiency while maintaining image quality. **Limitations:** Although our approach achieves efficient diffusion sampling, it faces challenges in generating quality when using a limited number of sampling steps (e.g., 5). In addition, even though our proposed parallelization can also be applied to network distillation approaches [18, 17, 19], we have not explored this direction in this paper and leave it to future research.

## Acknowledgements

This work was supported by NSFC (NO. 62225604) and Youth Foundation (62202243). We acknowledge project PID2022-143257NB-I00, financed by the Spanish Government MCIN/AEI/10.13039/501100011033 and FEDER. We acknowledge "Science and Technology Yongjiang 2035" key technology breakthrough plan project (2024Z120). Computation is supported by the Supercomputing Center of Nankai University (NKSC).

We would like to thank Kai Wang, a postdoctoral researcher at the Computer Vision Center, Universitat Autònoma de Barcelona, for his helpful discussions and comments during the rebuttal period.

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

# Contents

## Appendix

In the appendix, we provide a detailed description of the experimental implementation (Appendix A). Subsequently, an analysis of the parameter quantities for the encoder and decoder is conducted (Appendix B).

Following that, we explain the methodology used to compute Clipscore (Appendix C), as well as highlight the distinction between our approach and methods that reduce sampling steps (Appendix D). Then, we present additional experiments and results, including more detailed ablation studies and comparative experiments (Appendix F). In the end, we provide a statement of the potential broader impact of our work (Appendix H).

## A  Implementation Details

As shown in Code. 1, in the standard SD sampling code, adding 3 lines of code with green comments can achieve encoder propagation.

Code 1: Encoder propagation for SD (DDIM)

```
from diffusers import StableDiffusionPipeline
import torch
from utils import register_parallel_pipeline, register_faster_forward  # 1.import package

model_id = "runwayml/stable-diffusion-v1-5"
pipe = StableDiffusionPipeline.from_pretrained(model_id, torch_dtype=torch.float16)
pipe = pipe.to("cuda")

register_parallel_pipeline(pipe) # 2. enable parallel
register_faster_forward(pipe.unet) # 3. encoder propagation

prompt = "a photo of an astronaut riding a horse on mars"
image = pipe(prompt).images[0]
image.save("astronaut_rides_horse.png")
```

### A.1  Configure

We use the Stable Diffuison v1.5 pre-trained model [8] and DeepFloyd-IF [9]. All of our inference experiments are conducted using an A40 GPU (48GB of VRAM).

We randomly select 100 captions from the MS-COCO 2017 validation dataset [43] as prompts for generating images. The analytical results presented in Sec. 3.2 are based on the statistical outcomes derived from these 100 generated images.

### A.2  Details about the layers in the UNet

The UNet in Stable Diffusion (SD) consists of an encoder $\mathcal{E}$, a bottleneck $\mathcal{B}$, and a decoder $\mathcal{D}$, respectively. We divide the UNet into specific blocks: $\mathcal{E} = \{\mathcal{E}(\cdot)_s\}$, $\mathcal{B} = \{\mathcal{B}(\cdot)_8\}$, and $\mathcal{D} = \{\mathcal{D}(\cdot)_s\}$, where $s \in \{8, 16, 32, 64\}$. $\mathcal{E}(\cdot)_s$ and $\mathcal{D}(\cdot)_s$ represent the block layers with input resolution $s$ in the encoder and decoder, respectively. Tab. 7 presents detailed information about the block architecture. Fig. 10 illustrates the hierarchical features for these blocks.

### A.3  Details about the blocks in the DiT

---

[8] https://huggingface.co/runwayml/stable-diffusion-v1-5
[9] https://github.com/deep-floyd/IF

Table 7: Detailed information about the layers of the encoder $\mathcal{E}$, bottleneck $\mathcal{B}$ and decoder $\mathcal{D}$ in the UNet of SD.

| UNet | | Layer number | Type of layer | Layer name | Input resolution of layer | Output resolution of layer |
|---|---|---|---|---|---|---|
| $\mathcal{E}$ | $\mathcal{E}(\cdot)_{64}$ | 0 | resnets | down_blocks.0.resnets.0 | (320, **64**, **64**) | (320, 64, 64) |
| | | 1 | attention | down_blocks.0.attentions.0 | (320, 64, 64) | (320, 64, 64) |
| | | 2 | resnet | down_blocks.0.resnets.1 | (320, 64, 64) | (320, 64, 64) |
| | | 3 | attention | down_blocks.0.attentions.1 | (320, 64, 64) | (320, 64, 64) |
| | | 4 | downsamplers | down_blocks.0.downsamplers.0 | (320, 64, 64) | (320, 32, 32) |
| | $\mathcal{E}(\cdot)_{32}$ | 5 | resnet | down_blocks.1.resnets.0 | (320, **32**, **32**) | (640, 32, 32) |
| | | 6 | attention | down_blocks.1.attentions.0 | (640, 32, 32) | (640, 32, 32) |
| | | 7 | resnet | down_blocks.1.resnets.1 | (640, 32, 32) | (640, 32, 32) |
| | | 8 | attention | down_blocks.1.attentions.1 | (640, 32, 32) | (640, 32, 32) |
| | | 9 | downsamplers | down_blocks.1.downsamplers.0 | (640, 32, 32) | (640, 16, 16) |
| | $\mathcal{E}(\cdot)_{16}$ | 10 | resnet | down_blocks.2.resnets.0 | (640, **16**, **16**) | (1280, 16, 16) |
| | | 11 | attention | down_blocks.2.attentions.0 | (1280, 16, 16) | (1280, 16, 16) |
| | | 12 | resnet | down_blocks.2.resnets.1 | (1280, 16, 16) | (1280, 16, 16) |
| | | 13 | attention | down_blocks.2.attentions.1 | (1280, 16, 16) | (1280, 16, 16) |
| | | 14 | downsamplers | down_blocks.2.downsamplers.0 | (1280, 16, 16) | (1280, 8, 8) |
| | $\mathcal{E}(\cdot)_{8}$ | 15 | resnet | down_blocks.3.resnets.0 | (1280, **8**, **8**) | (1280, 8, 8) |
| | | 16 | resnet | down_blocks.3.resnets.1 | (1280, 8, 8) | (1280, 8, 8) |
| $\mathcal{B}$ | $\mathcal{B}(\cdot)_{8}$ | 17 | resnet | mid_blocks.resnets.0 | (1280, **8**, **8**) | (1280, 8, 8) |
| | | 18 | attention | mid_blocks.attentions.0 | (1280, 8, 8) | (1280, 8, 8) |
| | | 19 | resnet | mid_blocks.resnets.1 | (1280, 8, 8) | (1280, 8, 8) |
| $\mathcal{D}$ | $\mathcal{D}(\cdot)_{8}$ | 20 | resnet | up_blocks.0.resnets.0 | (1280+1280, **8**, **8**) | (1280, 8, 8) |
| | | 21 | resnet | up_blocks.0.resnets.1 | (1280+1280, 8, 8) | (1280, 8, 8) |
| | | 22 | resnet | up_blocks.0.resnets.2 | (1280+1280, 8, 8) | (1280, 8, 8) |
| | | 23 | upsamplers | up_blocks.0.upsamplers.0 | (1280, 8, 8) | (1280, 16, 16) |
| | $\mathcal{D}(\cdot)_{16}$ | 24 | resnet | up_blocks.1.resnets.0 | (1280+1280, **16**, **16**) | (1280, 16, 16) |
| | | 25 | attention | up_blocks.1.attentions.0 | (1280, 16, 16) | (1280, 16, 16) |
| | | 26 | resnet | up_blocks.1.resnets.1 | (1280+1280, 16, 16) | (1280, 16, 16) |
| | | 27 | attention | up_blocks.1.attentions.1 | (1280, 16, 16) | (1280, 16, 16) |
| | | 28 | resnet | up_blocks.1.resnets.2 | (1280+640, 16, 16) | (1280, 16, 16) |
| | | 29 | attention | up_blocks.1.attentions.2 | (1280, 16, 16) | (1280, 16, 16) |
| | | 30 | upsamplers | up_blocks.1.upsamplers.0 | (1280, 16, 16) | (1280, 32, 32) |
| | $\mathcal{D}(\cdot)_{32}$ | 31 | resnet | up_blocks.2.resnets.0 | (1280+640, **32**, **32**) | (640, 32, 32) |
| | | 32 | attention | up_blocks.2.attentions.0 | (640, 32, 32) | (640, 32, 32) |
| | | 33 | resnet | up_blocks.2.resnets.1 | (640+640, 32, 32) | (640, 32, 32) |
| | | 34 | attention | up_blocks.2.attentions.1 | (640, 32, 32) | (640, 32, 32) |
| | | 35 | resnet | up_blocks.2.resnets.2 | (640+320, 32, 32) | (640, 32, 32) |
| | | 36 | attention | up_blocks.2.attentions.2 | (640, 32, 32) | (640, 32, 32) |
| | | 37 | upsamplers | up_blocks.2.upsamplers.0 | (640, 32, 32) | (640, 64, 64) |
| | $\mathcal{D}(\cdot)_{64}$ | 38 | resnet | up_blocks.3.resnets.0 | (640+320, **64**, **64**) | (320, 64, 64) |
| | | 39 | attention | up_blocks.3.attentions.0 | (320, 64, 64) | (320, 64, 64) |
| | | 40 | resnet | up_blocks.3.resnets.1 | (320+320, 64, 64) | (320, 64, 64) |
| | | 41 | attention | up_blocks.3.attentions.1 | (320, 64, 64) | (320, 64, 64) |
| | | 42 | resnet | up_blocks.3.resnets.2 | (320+320, 64, 64) | (320, 64, 64) |
| | | 43 | attention | up_blocks.3.attentions.2 | (320, 64, 64) | (320, 64, 64) |

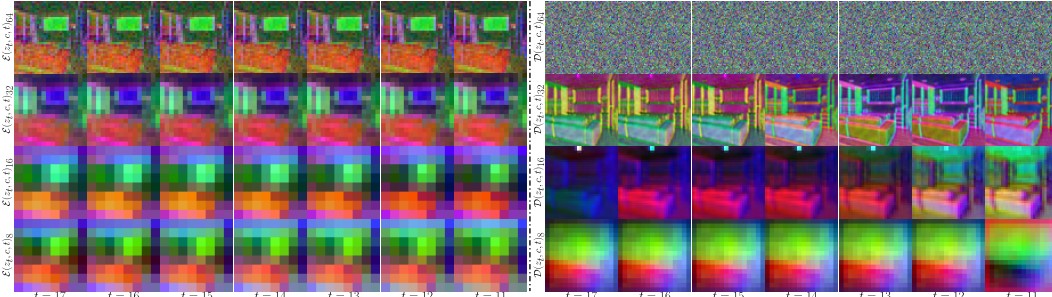

Figure 10: Visualising the hierarchical features. We applied PCA to the hierarchical features following PnP [20] and used the top three leading components as an RGB image for visualization. The encoder features changes minimally and has similarity at many time-steps (left), while the decoder features exhibit substantial variations across different time-steps (right).

Fig. 12 visualizes the hierarchical features for DiT [21] blocks, which includes 28 transformer blocks. Through visualization and statistical analysis (see Fig. 12 and Fig. 11), we observe that the features in the first several transformer blocks change minimally (i.e., the first 18 blocks), similar to the Encoder in SD, while the features in the remaining transformer blocks exhibit substantial variations (i.e., the remaining 10 blocks), akin to the Decoder in SD. For ease of presentation, we refer to the first several transformer blocks of DiT as the Encoder and the remaining transformer blocks as the Decoder. In Tab. 3, we demonstrate the accelerating performance of our method while applied to DiT-based generation models.

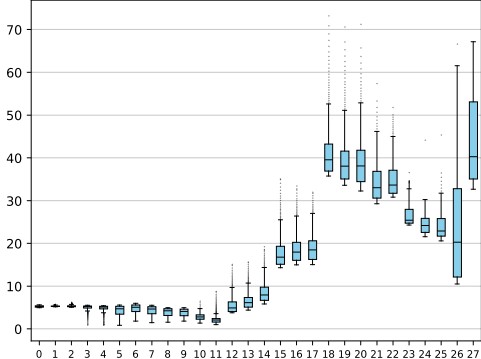

Figure 11: DiT feature statistics (F-norm)

## A.4 Time and memory consumption ratios

We report the run time and GPU memory consumption ratios for text-to-image task. As shown in Tab. 8, we significantly accelerate the diffusion sampling with encoder propagation while maintaining a comparable memory demand to the baselines (Tab. 8 (the last two columns)). Specifically, our proposed method reduces the spending time (s/image) by 24% and requires a little additional memory compared to standard DDIM sampling in SD (DDIM vs. Ours: 2.62GB vs. 2.64GB). The increased GPU memory requirement is for caching the features of the encoder from the previous time-step. Though applying parallel encoder propagation results in an increase in memory requirements by 51%, it leads to a more remarkable acceleration of 41% (DDIM vs. Ours: 2.62GB vs. 3.95GB). In conclusion, applying encoder propagation reduces the sampling time, accompanied by a negligible increase in memory requirements. Parallel encoder propagation on text-to-image tasks yields a sampling speed improvement of 20% to 43%, requiring an additional acceptable amount of memory.

Besides the standard text-to-image task, we also validate our proposed approach on other tasks: *text-to-video generation* (i.e., Text2Video-zero [4] and VideoFusion [5]), *personalized generation* (i.e., Dreambooth [7] and Custom Diffusion [8]) and *reference-guided image generation* (i.e., ControlNet [10]). We present the time and memory consumption ratios for these tasks in Tab. 9.

As reported in Tab. 9 (top), when combined with our method, there is a reduction in sampling time by 23% and 32% for Text2Video-zero [4] and VideoFusion [5], respectively, while the memory requirements increased slightly by 3% (0.2GB) and 0.9% (0.11GB).

The time spent by reference-guided image generation (i.e., ControlNet [10]) is reduced by more than 20% with a negligible increase in memory (1%). When integrated with our parallel encoder propagation, the sampling time in this task can be reduced by more than half (51%) (Tab. 9 (middle)). Dreambooth [7] and Custom Diffusion [8] are two approaches for customizing tasks by fine-tuning text-to-image diffusion models. As reported in Tab. 9 (bottom), our method, working in conjunction with the two customization approaches, accelerates the image generation with an acceptable increase in memory.

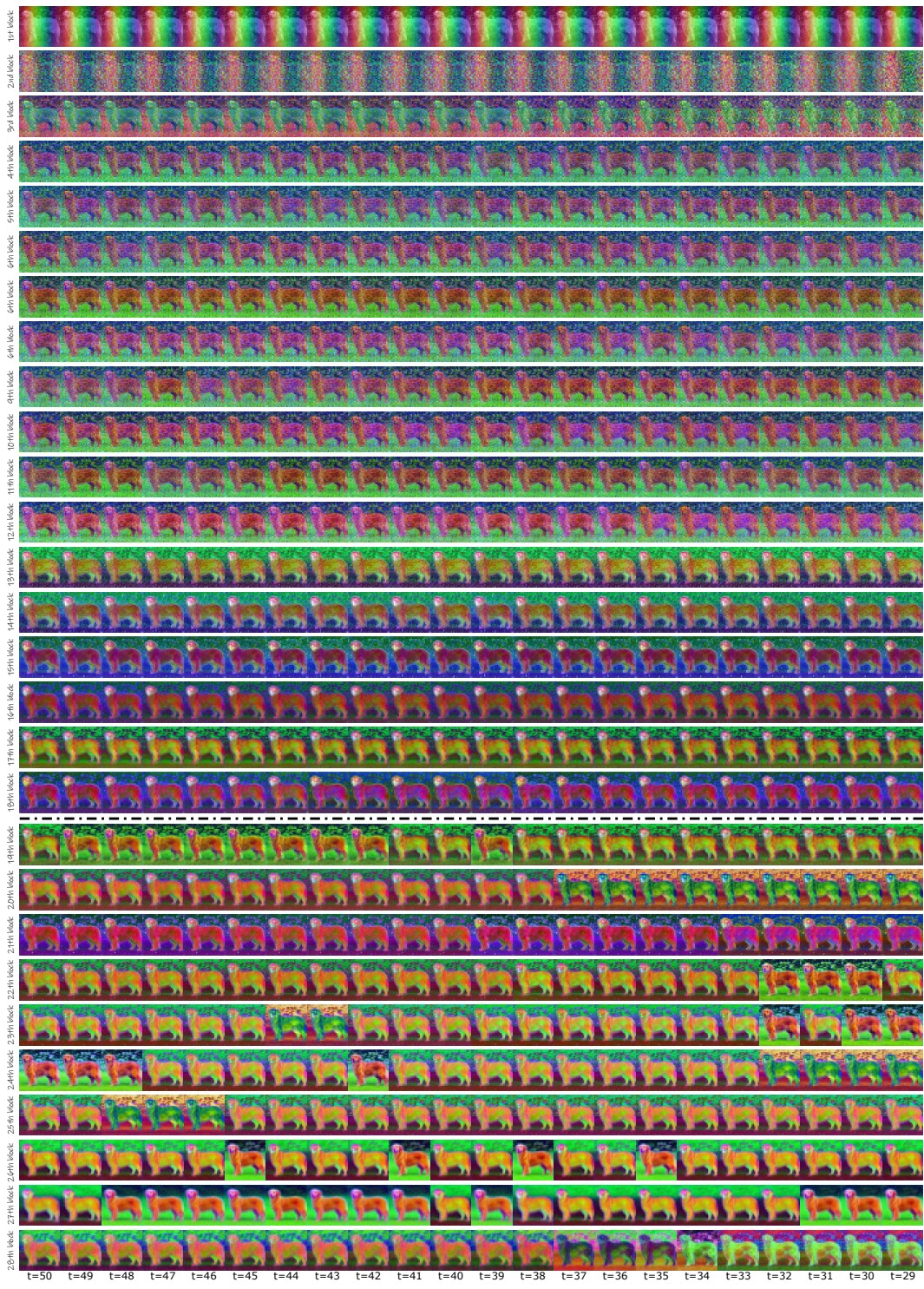

Figure 12: Visualising the hierarchical features. We apply PCA to the hierarchical features following PnP [20], and use the top three leading components as an RGB image for visualization. The output features of the first 18 blocks change slowly, and show similarity at many time steps (top), while the ones in the remaining 10 blocks exhibit substantial variations across different time steps (bottom).

Note that our method, either utilizing encoder propagation or parallel encoder propagation, improves sampling speed without compromising image quality (Sec. 4.1 and Sec. 4.3). We

Table 8: Time and GPU memory consumption ratios in both SD model and DeepFloyd-IF diffusion model. †: Encoder propagation, ‡: Parallel encoder propagation.

| DM | Sampling Method | | T | s/image | memory (GB) |
|---|---|---|---|---|---|
| Stable Diffusion | DDIM [13] | | | 2.42 | 2.62 |
| | DDIM [13] w/ Ours | † ‡ | 50 | **1.82**$_{24\%\downarrow}$ **1.42**$_{41\%\downarrow}$ | 2.64 3.95 |
| | DPM-Solver [14] | | | 1.14 | 2.62 |
| | DPM-Solver [14] w/ Ours | † ‡ | 20 | **0.92**$_{19\%\downarrow}$ **0.64**$_{43\%\downarrow}$ | 2.64 2.69 |
| | DPM-Solver++ [47] | | | 1.13 | 2.61 |
| | DPM-Solver++ [47] w/ Ours | † ‡ | 20 | **0.91**$_{19\%\downarrow}$ **0.64**$_{43\%\downarrow}$ | 2.65 2.68 |
| | DDIM + ToMe [35] | | 50 | 2.26 | 2.62 |
| | DDIM + ToMe [35] w/ Ours | † ‡ | | **1.72**$_{24\%\downarrow}$ **1.33**$_{41\%\downarrow}$ | 2.64 3.95 |
| DeepFloyd-IF | DDPM [2] | | | 34.55 | 40.5 |
| | DDPM [2] w/ Ours | † ‡ | 225 | **29.45**$_{15\%\downarrow}$ **26.27**$_{24\%\downarrow}$ | 41.1 41.1 |
| | DPM-Solver++ [47] | | | 16.09 | 40.5 |
| | DPM-Solver++ [47] w/ Ours | † ‡ | 100 | **14.13**$_{12\%\downarrow}$ **12.97**$_{20\%\downarrow}$ | 40.8 40.8 |

Table 9: Time and GPU memory consumption ratios in text-to-video, personalized generation and reference-guided generated tasks. †: Encoder propagation, ‡: Parallel encoder propagation.

| Sampling Method | | T | s/image | memory (GB) |
|---|---|---|---|---|
| Text2Video-zero [4] | | | 13.65 | 6.59 |
| Text2Video-zero [4] w/ Ours | † ‡ | 50 | **10.54**$_{23\%\downarrow}$ – | 6.79 – |
| VideoFusion [5] | | | 17.93 | 11.87 |
| VideoFusion [5] w/ Ours | † ‡ | 50 | **12.27**$_{32\%\downarrow}$ – | 11.98 – |
| ControlNet [10] (edges) | | | 3.20 | 3.81 |
| ControlNet [10] (edges) w/ Ours | † ‡ | 50 | **2.01**$_{37\%\downarrow}$ **1.52**$_{51\%\downarrow}$ | 3.85 5.09 |
| ControlNet [10] (scribble) | | | 3.95 | 3.53 |
| ControlNet [10] (scribble) w/ Ours | † ‡ | 50 | **3.18**$_{20\%\downarrow}$ **1.93**$_{51\%\downarrow}$ | 3.57 4.45 |
| Dreambooth [7] | | | 2.42 | 2.61 |
| Dreambooth [7] w/ Ours | † ‡ | 50 | **1.81**$_{24\%\downarrow}$ **1.42**$_{41\%\downarrow}$ | 2.65 3.93 |
| Custom Diffusion [8] | | | 2.42 | 2.61 |
| Custom Diffusion [8] w/ Ours | † ‡ | 50 | **1.82**$_{24\%\downarrow}$ **1.42**$_{41\%\downarrow}$ | 2.64 3.94 |

conducted GPU memory consumption ratios using the official method provided by PyTorch, `torch.cuda.max_memory_allocated` [10] , which records the peak allocated memory since the start of the program.

## A.5 GFLOPs

We use the fvcore[11] library to calculate the GFLOPs required for a single forward pass of the diffusion model. Multiplying this by the number of sampling steps gives us the total computational burden for sampling one image. Additionally, using fvcore, we can determine the computational load required for each layer of the diffusion model. Based on the key time-steps set in our experiments, we subtract the computation we save from the original total computational load, which then represents the GFLOPs required by our method. Similarly, fvcore also supports parameter count statistics.

## A.6 Baseline Implementations

For the comparisons of text-to-image generation in Sec. 4, we use the official implementation of DPM-Solver [14], DPM-Solver++ [47] [12], and ToMe [35] [13]. For the other tasks with text-guided diffusion model, we use the official implementation of Text2Video-zero [4] [14], VideoFusion [5] [15], ControlNet [10] [16], Dreamboth [7] [17], and Custom Diffusion [8] [18]. We maintain the original implementations of these baselines and directly integrate code into their existing implementations to implement our method.

---

[10] https://pytorch.org/docs/stable/generated/torch.cuda.max_memory_allocated.html

[11] https://github.com/facebookresearch/fvcore

[12] https://github.com/LuChengTHU/dpm-solver

[13] https://github.com/dbolya/tomesd

[14] https://github.com/Picsart-AI-Research/Text2Video-Zero

[15] https://huggingface.co/docs/diffusers/api/pipelines/text_to_video

[16] https://github.com/lllyasviel/ControlNet

[17] https://github.com/google/dreambooth

[18] https://github.com/adobe-research/custom-diffusion

Table 10: Model complexity comparison regarding the encoder $\mathcal{E}$, the bottleneck $\mathcal{B}$ and the decoder $\mathcal{D}$ in terms of parameter count and FLOPs.

|  | Parameter (billion) | FLOPs (million) |
|---|---|---|
| $\mathcal{E} + \mathcal{B}$ | $0.25 + 0.097$ | $224.2 + 6.04$ |
| $\mathcal{D}$ | $0.52_{1.47\times}$ | $504.4_{2.2\times}$ |

## B    Parameter Count and FLOPs of SD

We take into account model complexity in terms of parameter count and FLOPs (see Tab. 10). It's noteworthy that the decoder $\mathcal{D}$ exhibits a significantly greater parameter count, totaling 0.51 billion. This figure is approximately 1.47 times the number of parameter combinations for the encoder $\mathcal{E}$ (250 million) and the bottleneck $\mathcal{B}$ (97 million). This substantial parameter discrepancy suggests that the decoder $\mathcal{D}$ carries a more substantial load in terms of model complexity.

Furthermore, when we consider the computational load, during a single forward inference pass of the SD model, the decoder $\mathcal{D}$ incurs a considerable 504.4 million FLOPs. This value is notably higher, approximately 2.2 times, than the cumulative computational load of the encoder $\mathcal{E}$, which amounts to 224.2 million FLOPs, and the bottleneck $\mathcal{B}$, which requires 6.04 million FLOPs. This observation strongly implies that the decoder $\mathcal{D}$ plays a relatively more important role in processing and transforming the data within the UNet architecture, emphasizing its critical part in the overall functionality of the model.

## C    The methodology for computing Clipscore

Clipscore [46] is a metric for computing the consistency between text and image. The formula for computing Clip-score between image embedding $v$ and text embedding $c$, as presented in the original paper, is as follows:

$$\text{CLIP-S} = w * \max(\cos(c, v), 0),$$

where a re-scaling operation is performed using $w$, set to 2.5 in the official implementation of Clipscore [46]. The rationale behind the re-scaling operation, as provided by the official paper (excerpted from [46], Section 3, Footnote 6), is as follows:

> While the cosine similarity, in theory, can range from $[-1, 1]$ (1) we never observed a negative cosine similarity; and (2) we generally observe values ranging from roughly zero to roughly .4. The particular value of $w$ we advocate for, $w = 2.5$, attempts to stretch the range of the score distribution to $[0, 1]$.

Therefore, adhering to the aforementioned setting, we employed the official implementation of Clipscore for evaluation, yielding Clipscore data distributed around 0.75. Some works [8, 48] employ re-scaling operations during evaluation, yielding Clipscore values around 0.75, while others [10, 4] do not, resulting in Clipscore values around 0.3. In essence, both approaches are equivalent, differing only in whether re-scaling is applied at the end. We evaluate Clipscore using $w = 2.5$ for re-scaling, hence yielding Clipscore data around 0.75, unless otherwise stated.

## D    Different from step-reduction methods

Several efficient diffusion model solvers, such as DDIM [13], DPM-Solver [14] and DPM-Slover++ [47], have significantly reduced sampling steps. Our method does not reduce the number of sampling steps. In the encoder propagation, the decoder needs to compute for all time-steps, and we require time embedding inputs for all time-steps to maintain temporal coherence.

Fig. 13 illustrates the qualitative comparison results of SD (DDIM) showcasing a reduction in steps (i.e., 9 and 25 time-steps). We present the results with 9 steps because the number of *key* time-steps for the SD model with DDIM in our approach is 9. The generation quality notably declines with step reduction, attributed to alterations in image structure and a diminished attention to detail generation,

Two girls holding flowers and wearing sunglasses

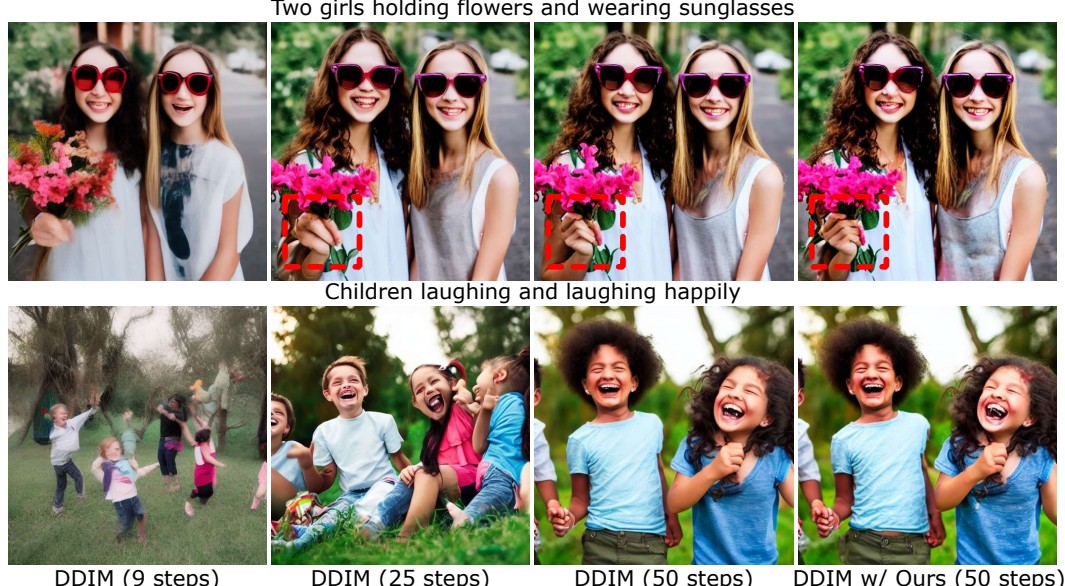

Children laughing and laughing happily

| DDIM (9 steps) | DDIM (25 steps) | DDIM (50 steps) | DDIM w/ Ours (50 steps) |

Figure 13: When lowering the time-steps in inference, the image quality noticeably deteriorates, while ours maintains a similar image quality to the original.

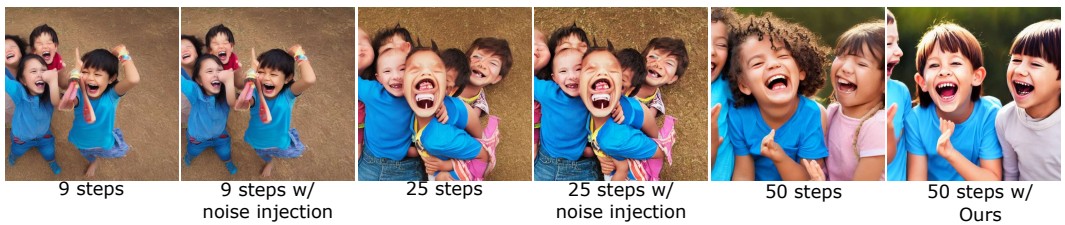

| 9 steps | 9 steps w/ noise injection | 25 steps | 25 steps w/ noise injection | 50 steps | 50 steps w/ Ours |

Figure 14: Fewer sampling steps with noise injection.

Table 11: Quantitative comparison for DDIM with fewer steps.

| Sampling method | T | FID ↓ | Clipscore ↑ | s/image↓ |
|---|---|---|---|---|
| DDIM | 50 | 21.75 | 0.773 | 2.42 |
| DDIM | 25 | 22.16 | 0.761 | 1.54 |
| DDIM w/ noise injection | 25 | 21.89 | 0.761 | 1.54 |
| DDIM | 9 | 27.58 | 0.735 | 0.96 |
| DDIM w/ noise injection | 9 | 27.63 | 0.736 | 0.96 |
| DDIM w/ ours | 50 | 21.08 | 0.783 | 1.42 |

Table 12: Quantitative comparison for DPM-Solver/DPM-solver++ with fewer steps.

| Sampling method | T | FID ↓ | Clipscore ↑ |
|---|---|---|---|
| DPM-Solver | 20 | 21.36 | 0.780 |
| DPM-Solver | 10 | 22.41 | 0.768 |
| DPM-Solver w/ ours | 20 | 21.25 | 0.779 |
| DPM-Solver++ | 20 | 20.51 | 0.782 |
| DPM-Solver++ | 10 | 21.25 | 0.771 |
| DPM-Solver++ w/ ours | 20 | 20.76 | 0.781 |

as exemplified by the **hands** in Fig. 13 (the second cloumn). We increase the number of sampling steps (i.e., 25 steps), and find that the sampling results do not perform as well as DDIM with 50 steps and its variation with FasterDiffusion (Fig. 13 and Tab. 11). As shown in Tab. 11, the FID and Clipscore of SD (DDIM) with 25 time-steps on the MS-COCO 2017 10K subset are 22.16 and 0.761, respectively, significantly worse than the results obtained with 50 time-steps and ours (FID↓: 21.75, 21.08; Clipscore↑: 0.773, 0.783). This demonstrates that our method is not simply reducing the number of sampling steps. With fewer steps (T=10), both DPM-Solver and DPM-Solver++ also

exhibit worse FID and Clipscore (see Tab. 12). These results indicate that our method achieves better performance compared to simply reducing the sampling steps.

As shown on Tab. 11, we conduct DDIM scheduler inference time with various steps. Although FasterDiffusion (the last row) is slightly longer than DDIM scheduler with 9 steps, our sampling results are much closer the 50-step DDIM generation quality, whereas the 9-step DDIM results are much inferior (see Tab. 11, Tab. 12, and Fig. 13 for examples).

For directly applying noise injection to the generation phase, we show image generation examples in Fig. 14 in the rebuttal file. Fewer sampling steps equipped with prior noise injection result in almost no improvement in image quality. Tab. 11 further supports this conclusion. While in our case, FasterDiffusion working without the noise injection will lead to smoother textures, as shown in Fig.6. The noise injection technique helps in preserving fidelity in the generated results.

# E    Difference between "prior noise injection" and "churn"

In EDM [49] (Karras et al.), they increase the noise level during ODE sampling to improve deterministic sampling, which is referred to as stochastic sampling (i.e., "churn"). Compared to deterministic sampling, it injects noise into the image at each step, which helps to enhance the quality of the generated images. Song et al. [50] first observed that perturbing data with random Gaussian noise makes the data distribution more amenable to score-based generative modeling. Increasing the noise level in EDM is based on Song's paper.

Their purpose for injecting noise is to perturb the data, whereas our purpose for injecting noise during sampling is to preserve high-frequency details in the image during the denoising process, preventing the diffusion model from removing high-frequency information as noise (see Figure.6 in the main paper). In addition, our method FasterDiffusion differs from them by only inserting noise in the later stage of the diffusion steps, while they insert noise to all time-steps.

# F    Ablation Experiments and Additional Results

## F.1    Comparion with DeepCache

DeepCache is developed based on the observation over the temporal consistency between high-level features. In this paper, we have a more thorough analytical study over the SD model features as shown in Fig.3, where we find out the encoder features change less, whereas the decoder features exhibit substantial variations across different time steps. This insight motivates us to omit encoder computation at certain adjacent time-steps and reuse encoder features of previous time-steps as input to the decoder in multiple time-steps. We further determine the key time steps in the T2I inference stage, which helps our method to skip the time steps in a more scientific way.

It is also important to note that DeepCache is not parallelizable on multiple GPUs since DeepCache needs to use all or part of the encoder and decoder at every time step. FasterDiffusion, on the other hand, only uses the encoder at the *key* time steps, which enables parallel processing at these time steps and faster inference time cost.

It is also worth to notice that FasterDiffusion can be further combined with a wide range of diffusion model based tasks, including Text2Video-zero, VideoFusion, Dreambooth, and ControlNet. In this case, DeepCache often shows much slower speed while applied to these tasks. As an example shown in Table. 13, when combined with ControlNet DeepCache is slower by 24% compared to the FasterDiffusion.

More specifically, since the ControlNet model requires an additional encoder, our method FasterDiffusion is able to execute this extra encoder in a parallel manner and reuse it, and that makes the additional time negligible. On the other hand, DeepCache is reusing the decoder feature, which leads the ControlNet to wait for the additional encoder to complete computation at each time-step, resulting in almost no time saved from skipping the encoder in the standard SD models. The T2I generation results are shown in Figure. 15, FasterDiffusion successfully preserves the given structure information and achieves similar results as the original ControlNet model.

Table 13: When combined with ControlNet (Edge) 50-step DDIM, our inference time shows a significant advantage compared to DeepCache.

|  | Clipscore↑ | FID↓ | s/image↓ |
|---|---|---|---|
| ContrlNet | 0.769 | 13.78 | 3.20 |
| ContrlNet w/ DeepCache | 0.765 | 14.18 | 1.89 (1.69x) |
| ContrlNet w/ Ours | 0.767 | 14.65 | **1.52 (2.10x)** |

| Input image | Canny condition | Stable diffusion | DeepCache | Ours |
|---|---|---|---|---|

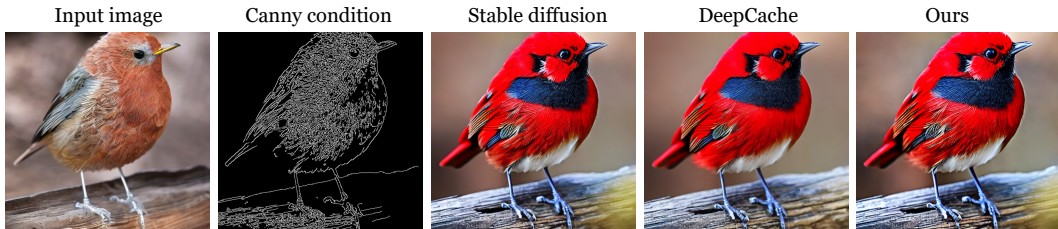

Figure 15: Combine our method with ControlNet.

### F.2 The definition of *key* time-steps in various tasks

We utilize 50, 20 and 20 time-steps for DDIM scheduler [13], DPM-solver [14] and DPM-solver++ [47], respectively. We follow the official implementation of DeepFloyd-IF, setting the time-steps for the three sampling stages to 100, 50, and 75, respectively.

**Text-to-image generation.** We experimentally define the *key* time-steps as $t^{key} = \{50, 49, 48, 47, 45, 40, 35, 25, 15\}$ for SD model with DDIM [13], and $t^{key} = \{100, 99, 98, \ldots, 92, 91, 90, 85, 80, \ldots, 25, 20, 15, 14, 13, \ldots, 2, 1\}$, $\{50, 49, \ldots, 2, 1\}$ and $\{75, 73, 70, 66, 61, 55, 48, 40, 31, 21, 10\}$ for three stages of DeepFloyd-IF with DDPM [2]. For SD with both DPM-Solver [14] and DPM-Solver++ [47], we experimentally set the *key* time-steps to $t^{key} = \{20, 19, 18, 17, 15, 10, 5\}$.

**Other tasks with text-guided diffusion model.** In addition to standard text-to-image tasks, we further validate our approach on other tasks with text-guided diffusion model (Sec. 4.2). These tasks are all based on the SD implementation of DDIM [13]. Through experiments, we set the *key* time-steps to $t^{key} = \{50, 49, 48, 47, 45, 40, 35, 25, 15\}$ for Text2Video-zero [4], VideoFusion [5], and ControlNet [10]. For personalized tasks (i.e., Dreambooth [7] and Custom Diffusion [8]), we set the *key* time steps to $t^{key} = \{50, 49, 48, 47, 45, 40, 35, 25, 15, 10\}$.

### F.3 The effectiveness of encoder propagation

In Sec. 3.2, we have demonstrated that encoder propagation (Fig. 16c) can preserve semantic consistency with standard SD sampling (Fig. 16a). However, images generated through decoder propagation often fail to match certain specific objects mentioned in the text prompt (Fig. 16d and Tab. 14 (the third row)).

We extended this strategy to include encoder and decoder propagation (Fig. 16e) as well as decoder and encoder dropping (Fig. 16f). Similarly, encoder and decoder propagation often fail to cover specific objects mentioned in the text prompt, leading to a degradation in the quality of the generated results (Fig. 16e and Tab. 14 (the fourth row)). On the other hand, decoder and encoder dropping are unable to completely denoise, resulting in the generation of images with noise (Fig. 16f and Tab. 14 (the fifth row)). Note that decoder and encoder dropping (Fig. 16f) is different from directly reducing the number of time-steps. Time embedding is required as input for each time-step, even when using only the encoder or decoder at each time-step.

### F.4 User study details

The study participants were volunteers from our college. The questionnaire consisted of 35 questions, each presenting two images: one from the baseline methods (including ControlNet, SD, DeepFloyd-

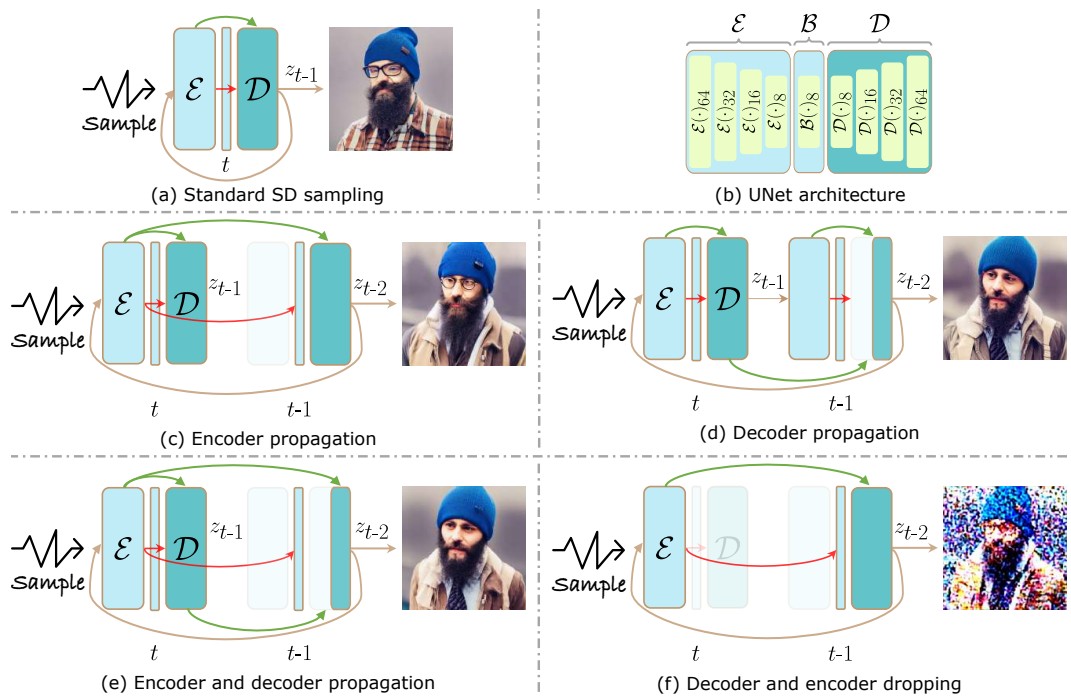

Figure 16: (a) Standard SD sampling. (b) UNet architecture. (c) Encoder propagation. (d) Decoder propagation. (e) Encoder and decoder propagation. (f) Decoder and encoder dropping.

Table 14: Quantitative evaluation for additional strategy on MS-COCO 2017 10K subset. Other propagation strategies can lead to the loss of some semantics under prompts and degradation of image quality (the third to fifth rows).

| Sampling method | T | FID ↓ | Clipscore ↑ |
|---|---|---|---|
| DDIM | 50 | 21.75 | 0.773 |
| DDIM w/ Encoder propagation (Ours) | 50 | **21.08** | **0.783** |
| DDIM w/ Decoder propagation | 50 | 22.97 | 0.758 |
| DDIM w/ Encoder and decoder propagation | 50 | 23.69 | 0.742 |
| DDIM w/ Decoder and encoder dropping | 50 | 199.48 | 0.679 |

IF, DPM-Solver++, Dreambooth, Custom Diffusion, Text-to-Video, etc.) and the other from our method FasterDiffusion (one example shown in Fig. 17).

Users were required to select the image where the target was more accurately portrayed, or choose "both equally good". A total of 18 users participated the questionnaire, resulting in totally 630 samples (35 questions × 1 option × 18 users). As the final results shown in Fig.8, the chosen percentages for SD, DeepFloyd-IF, DPM-Solver++, Text-to-Video, ControlNet, and Personalize were 48%, 44%, 51%, 52%, 46%, and 39%, respectively. These results show that our method performs on par with the baseline methods and demonstrate that FasterDiffusion retains the T2I generation quality while reducing the inference time.

## F.5 Additional metrics

We showcase our experimental results with the ImageReward [51] and PickScore [52] evaluation metrics over the MS-COCO2017 10K dataset, as shown in the Tab. 15. Our method FasterDiffusion enhances sampling efficiency while preserving the original model performance.

\* **12** Please select a high-quality image that matches the specified prompt.

\* **30** Please select a high-quality video that matches the specified prompt.

a cozy cabin nestled in a snowy mountain landscape

The tide is coming in on the sandy beach

A
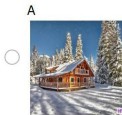
B
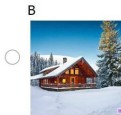
○ Option A and Option B have no significant differences.

A
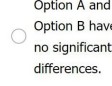
B
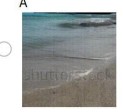
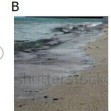
○ Option A and Option B have no significant differences.

Figure 17: User study examples.

|  | ImageReward↑ | PickScore↑ | s/image |
|---|---|---|---|
| SD (DDIM) | 0.149 | 52.10% | 2.42 |
| SD (DDIM) w/ Ours | 0.162 | 47.90% | 1.42 |

Table 15: Metrics in ImageReward and PickScore.

## F.6 Additional results

Fig. 18 shows additional results generated by DiT. In Figs. 19, 20 and 21, we show additional text-to-image generation results. Further results regarding other tasks with text-guided diffusion model are illustrated in Figs. 22, 23 and 24.

## F.7 Additional tasks

**ReVersion** [53] is a relation inversion method that relies on the Stable Diffusion (SD). Our method retains the capacity to generate images with specific relations based on exemplar images, as illustrated in Fig. 25 (top).

**P2P** [27] is an image editing method guided solely by text. When combined with our approach, it enhances sampling efficiency while preserving the editing effect (Fig. 25 (bottom)).

## G Automatic selection for key time steps

To identify key time steps, we conducted empirical analysis of feature changes at adjacent time steps in multistep Stable Diffusion models (50-step model as an example). This analysis is based on statistics from the distribution of features across 100 random prompts, which does not impose a high time cost. We found that the encoder features change minimally in later time steps, whereas the encoder features in earlier time steps exhibit substantial variations compared to later ones. Based on analysis as shown in Section 3.2, we determine the *key* time-step as $t^{key} = \{50, 49, 48, 47, 45, 40, 35, 25, 15\}$ for the 50-step Stable Diffusion model. As a general case, we also applied this *key* time-step configuration to the ControlNet [10], Text2Video-zero [4], VideoFusion [5], Dreambooth [7] and Custom Diffusion [8] models based on the 50-step SD models. That will not impose any additional searching time for the key time steps for these downstream application scenarios.

Existing methods such as OMS-DPM [54], AutoDiffusion [55], and DDSM [56] utilize reinforcement learning or machine learning algorithms to search for optimal schedules, time steps, and model sizes. Each search iteration involves frequent image generation and FID calculations. Executing these search algorithms is time-consuming, often exceeding the training time, as noted in DDSM. For instance, the NSGA-II search algorithm used in DDSM incurs a search cost approximately 1.1 to 1.2 times that of training a standard diffusion model. We plan to incorporate the NSGA-II algorithm for automatically searching $t^{key}$ time steps as an enhancement to FasterDiffusion in our future research.

## H Impact Statements

Diffusion models generate realistic fake images, which assist individuals in their creative endeavors. However, for those lacking the ability to discern whether the images are generated by a model, it may affect their judgment.

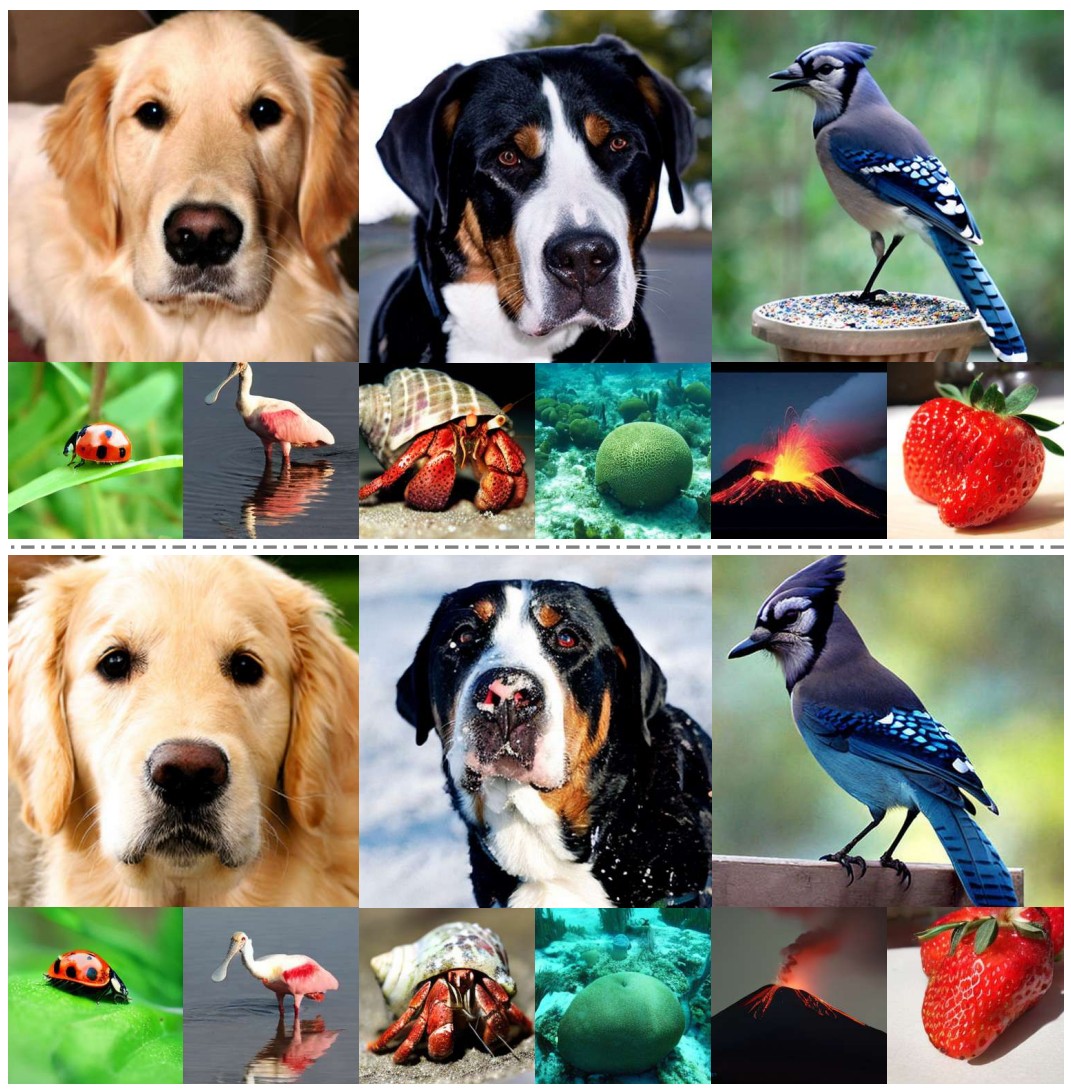

Figure 18: Additional results of DiT (top) and this method in conjunction with our proposed approach (bottom).

Equipped with our technique, using DMs on the edge devices becomes feasible. Also, importantly, our proposed acceleration of DMs does not require access to enormous compute, which is required to the methods based on knowledge distillation [18, 17, 19]. This makes our technique also useful for the many smaller actors in the generative AI field.

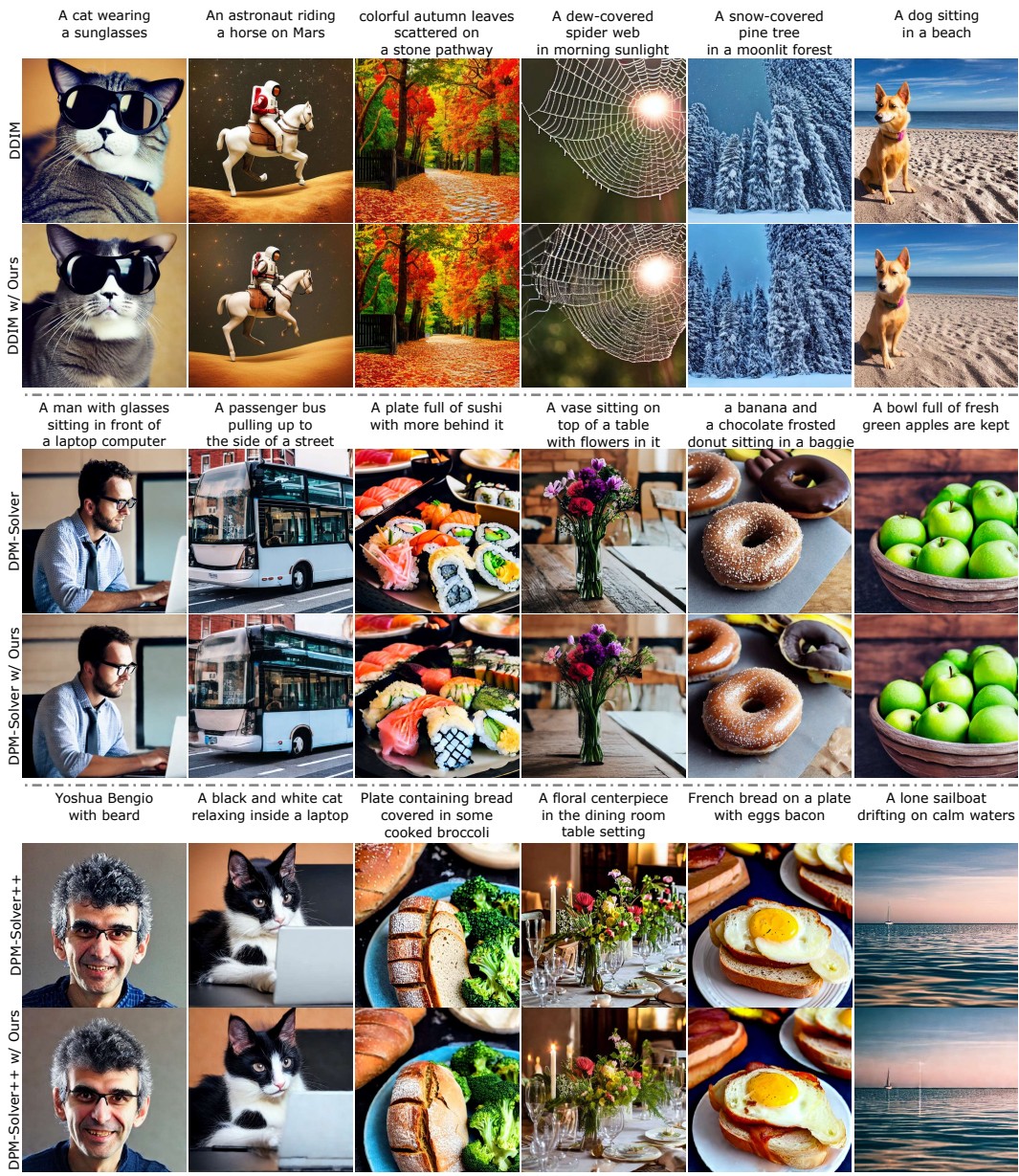

Figure 19: Additional results of text-to-image generation combining SD with DDIM [13], DPM-Solver [14], DPM-Solver++ [47], and these methods in conjunction with our proposed approach.

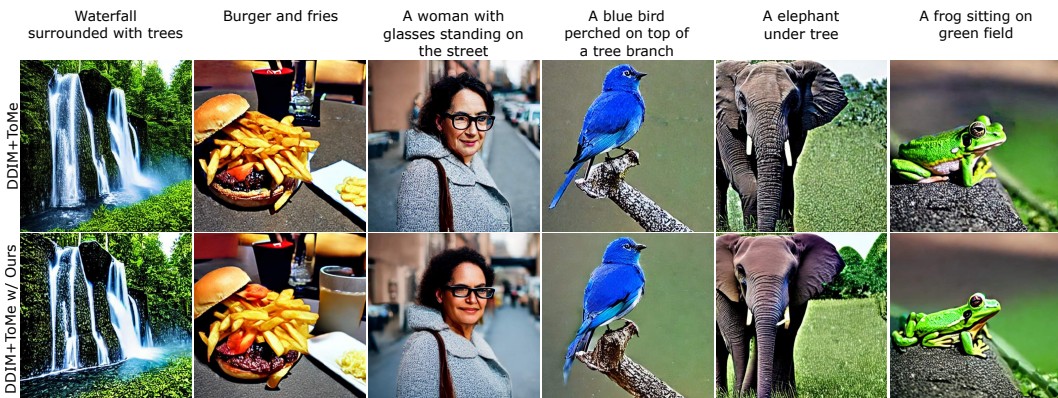

Figure 20: Additional results of text-to-image generation combining SD with DDIM+ToMe [35], and this method in conjunction with our proposed approach.

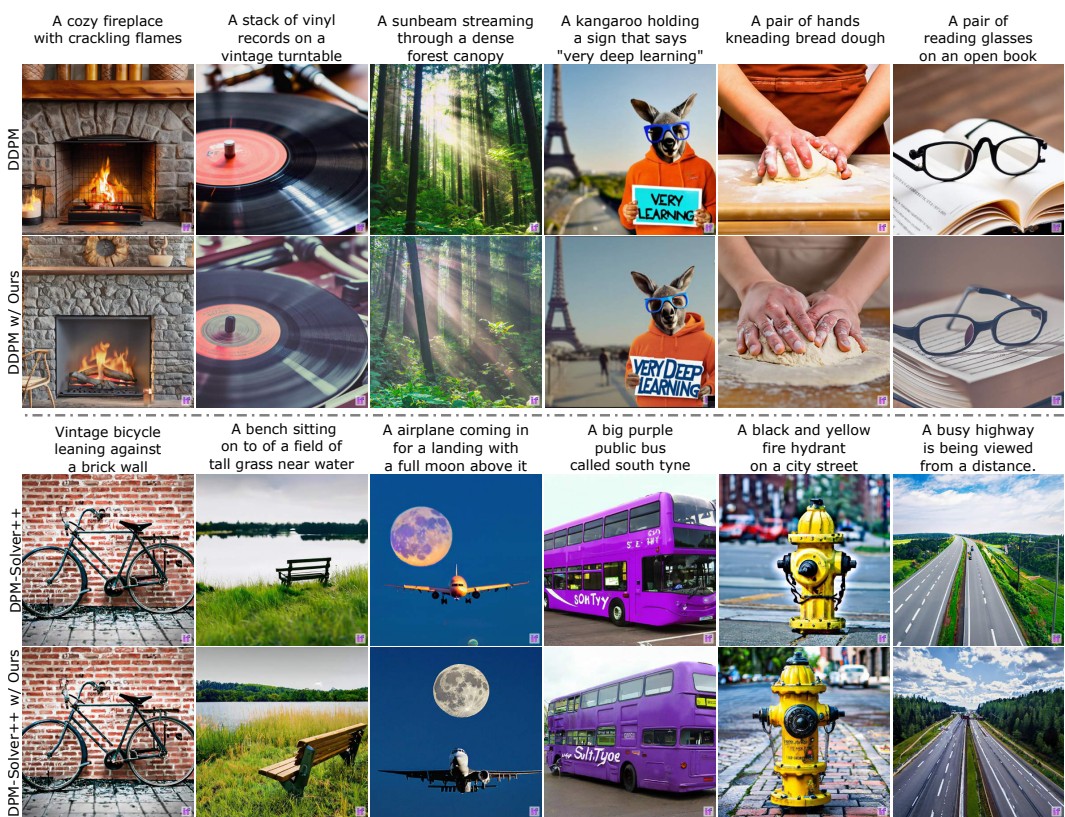

Figure 21: Additional results of text-to-image generation combining DeepFloyd-IF with DDPM [2] and DPM-Solver++ [47], and these methods in conjunction with our proposed approach.

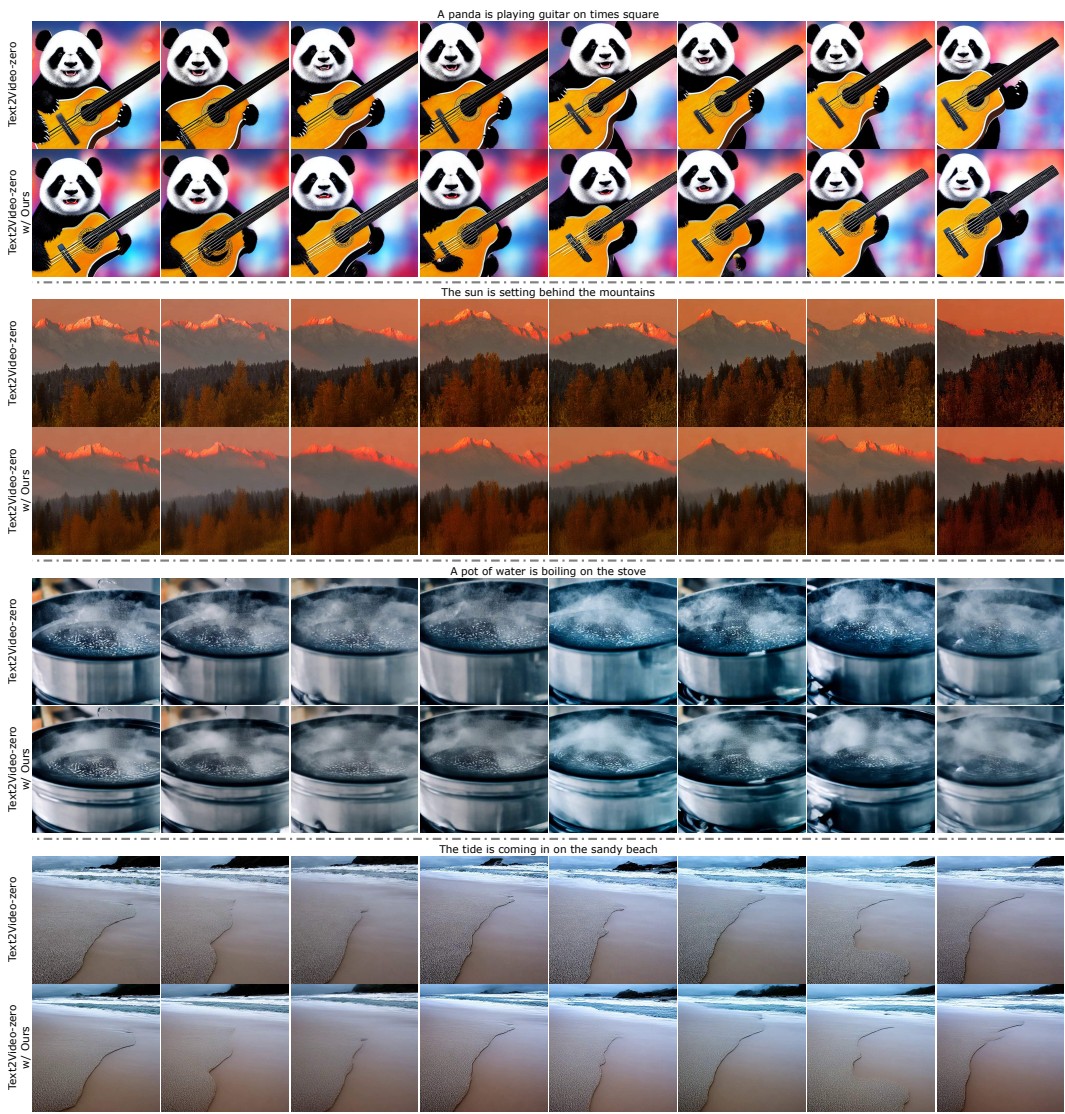

Figure 22: Additional results of Text2Video-zero [4] both independently and when combined with our proposed method.

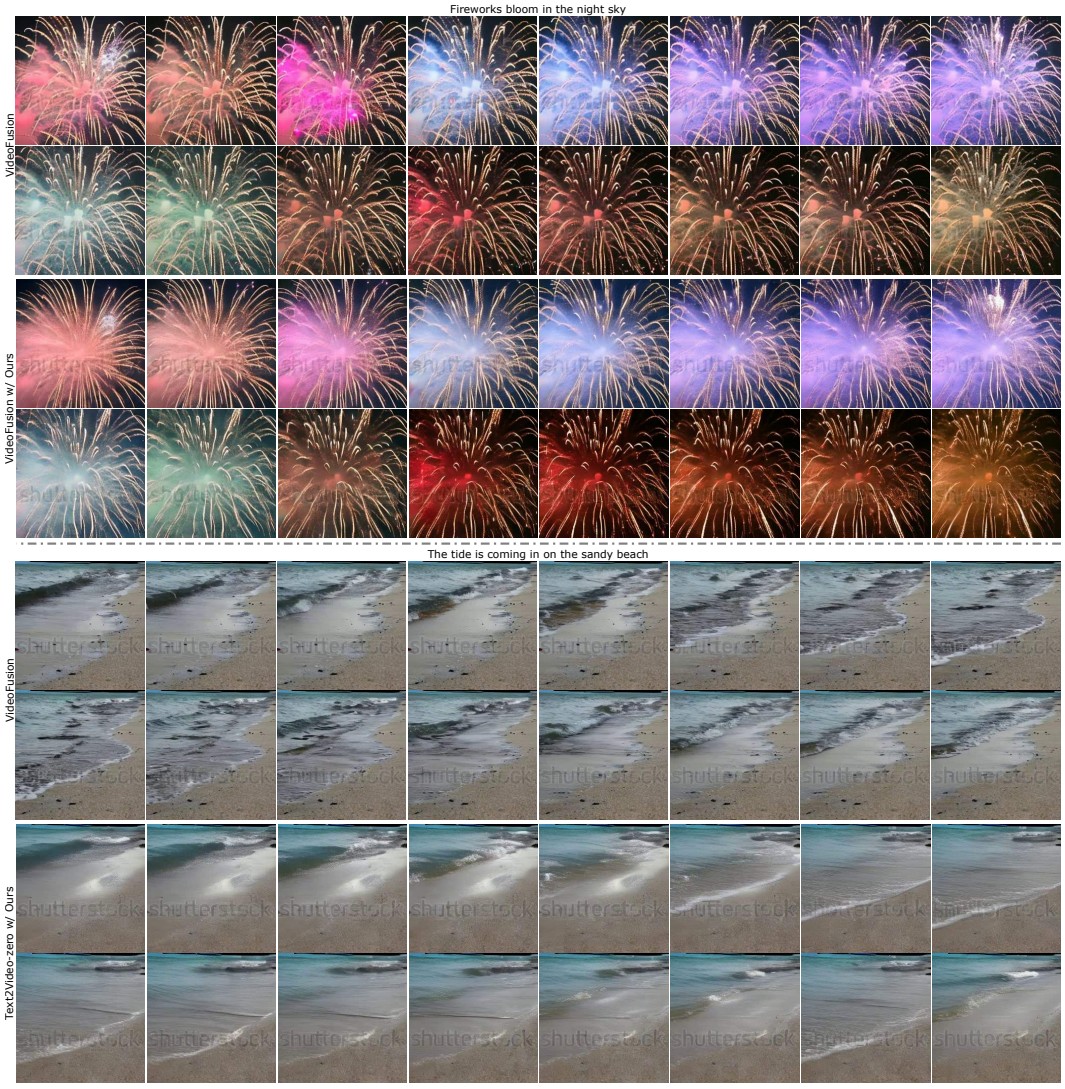

Figure 23: Additional results of VideoFusion [5] both independently and when combined with our proposed method.

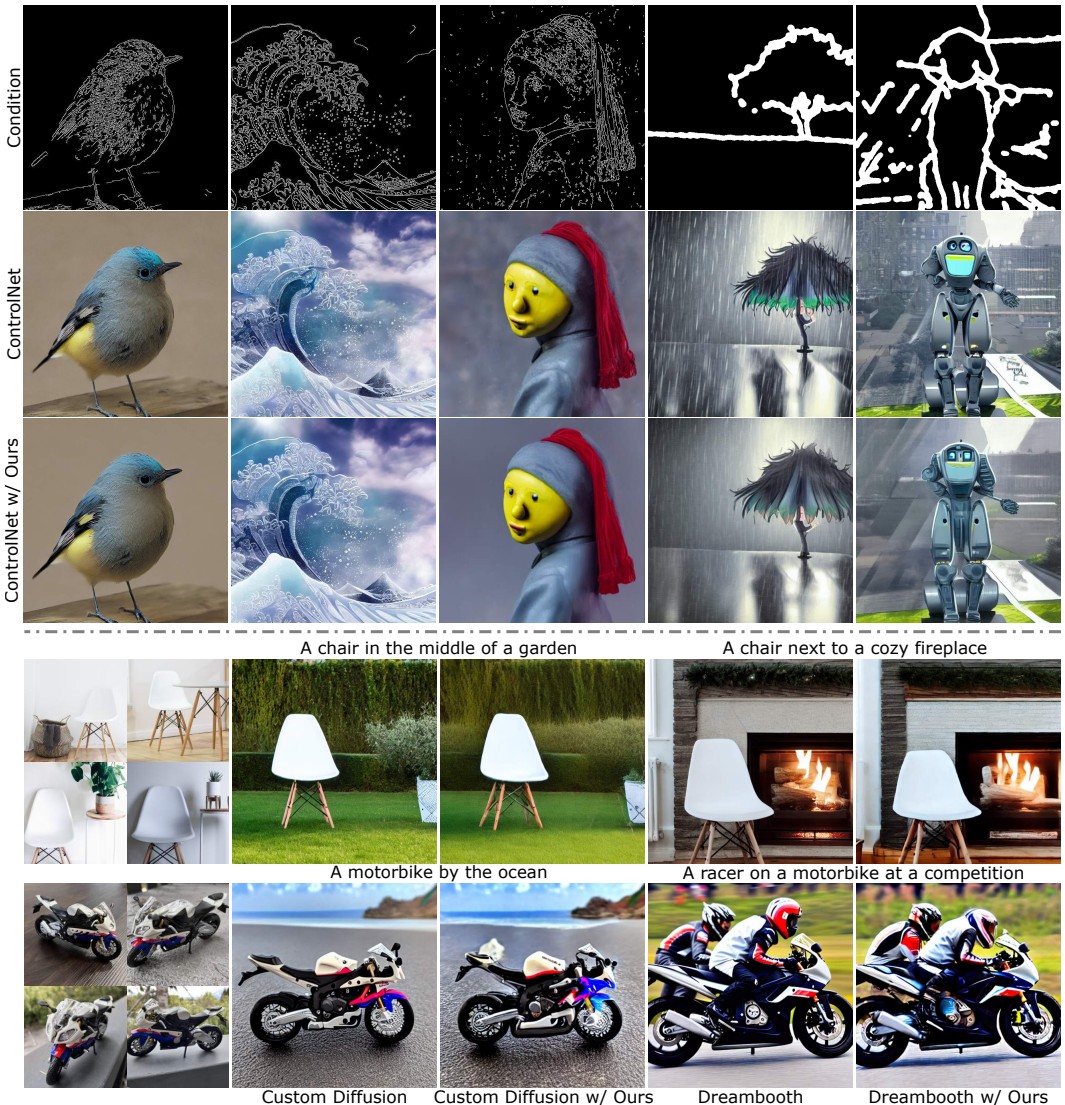

Figure 24: Additional results of ControlNet [10] (top) and personalized tasks [7, 8] (bottom) obtained both independently and in conjunction with our proposed method..

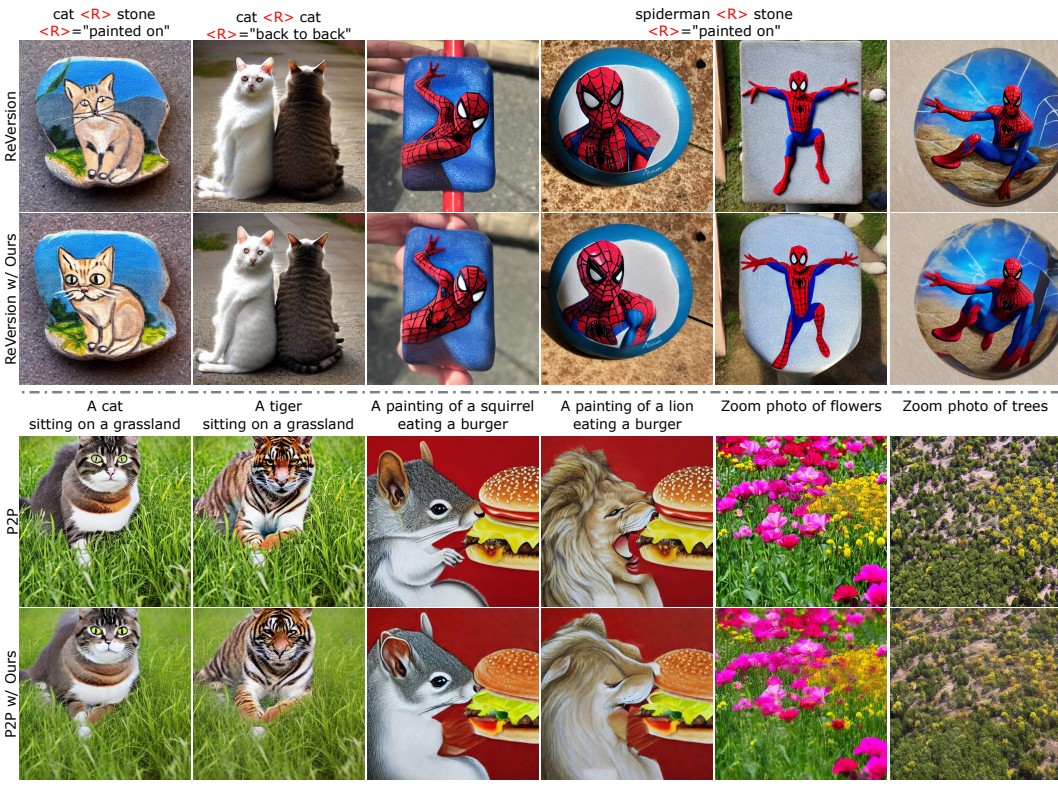

Figure 25: Edited results of ReVersion [53] (top) and P2P [27] (bottom) obtained both independently and in conjunction with our proposed method.

