# OpenReview forum: "Faster Diffusion: Rethinking the Role of the Encoder for Diffusion Model Inference"
_NeurIPS.cc/2024/Conference — NeurIPS 2024 poster_

### Official Review · Reviewer_i4aK · 2024-07-05

**Soundness:** 4
**Presentation:** 3
**Contribution:** 3
**Rating:** 8
**Confidence:** 4

**Summary:**

The authors analyze the encoder features of diffusion UNet and find that they share a lot of similarities across certain time steps. Based on this observation, the authors propose to reuse the encoder features and do parallel computation in sampling. This significantly fastens the generation process. Additionally, they also introduce a prior noise injection method to improve the generation quality. Extended experiments demonstrate their effectiveness across a wide range of generation tasks, including T2I, T2V, DreamBooth, controlNet. And their method can be seamlessly combined with various noise schedulers.

**Strengths:**

1 This paper is well-written and easy to understand. The analysis in section 1 is insightful. Fig.4 clearly shows the encoder propagation methods.
2 The empirical evaluation is thorough. The authors validate their method on various datasets, architectures, and use various metrics. They also make a lot of comparison to the SOTA acceleration methods, such as DeepCache.
3 Sec. A4 shows that the parallel denoising leads to an acceptable memory cost, which addresses my concerns.
4 This approach can be seamlessly combined with other acceleration methods such as novel noise schedulers and distillations.

Overall, this is an excellent work that significantly reduces diffusion sampling cost in a training-free manner.

**Weaknesses:**

I think the manual selection of key time-steps may not be optimal. Is it possible to use some automatic strategies to determine the key time-steps in your work? Otherwise when we use a new architecture we would have to redo the encoder feature analysis experiment and manually define the key-time steps again. I think there are several works have explored the selection of key time-steps. Discuss on them will make your work even stronger.

[1] OMS-DPM: Optimizing the Model Schedule for Diffusion Probabilistic Models
[2] AutoDiffusion: Training-Free Optimization of Time Steps and Architectures for Automated Diffusion Model Acceleration
[3] Denoising Diffusion Step-aware Models

**Questions:**

Please see the weakness section.

**Limitations:**

Yes, they discuss their limitation in section 5.

---

> ### Author Rebuttal · Authors · 2024-08-07
>
> We appreciate your feedback and will incorporate the discussions mentioned below to enhance the quality of our paper. Note that we utilize the numerical references to cite sources within the main paper.
>
> **W1. Automatic selection for key time steps**
>
> Thank you for your insightful suggestion. We find that when using uniform $\textit{key}$ time-steps with the same time interval, the sampling results are acceptable ($t^{key} = \\{50, 44, 38, 32, 26, 20, 14, 8, 2\\}$), though not as optimal compared to those obtained with non-uniform time-steps (see Table.5 in main paper).
>
> To identify key time steps, we conducted empirical analysis of feature changes at adjacent time steps in multistep Stable Diffusion models (50-step model as an example).
> This analysis is based on statistics from the distribution of features across 100 random prompts, which does not impose a high time cost.
> We found that the encoder features change minimally in later time steps, whereas the encoder features in earlier time steps exhibit substantial variations compared to later ones.
> Based on analysis as shown in Section 3.2, we determine the $\textit{key}$ time-step as $t^{key}=\\{50, 49, 48, 47, 45, 40, 35, 25, 15\\}$ for the 50-step Stable Diffusion model.
> As a general case, we also applied this $\textit{key}$ time-step configuration to the ControlNet [10], Text2Video-zero [4], VideoFusion [5], Dreambooth [7] and Custom Diffusion [8] models based on the 50-step SD models. That will not impose any additional searching time for the key time steps for these downstream application scenarios.
>
> As per your suggestions, we explored OMS-DPM [A], AutoDiffusion [B], and DDSM [C]. They are using reinforcement learning or machine learning algorithms to search for the optimal schedule, time steps, and model size, etc.
> Each search iteration involves high-frequent images generation and FID calculation.
> Executing these search algorithms is extensively time-consuming, often exceeding the training time, as claimed in DDSM.
> For example, the NSGA-II search algorithm is utilized in DDSM [C]. The search cost for the DDSM network is approximately 1.1 to 1.2 times that of training a standard diffusion model (**see the table below**).
> Due to the short rebuttal period, we have not been able to apply these algorithms to our method, however, we will discuss the use of the NSGA-II algorithm for automatically searching $t^{key}$ time-steps in the final version and include it as an improved version for FasterDiffusion in our future research.
>
> |                 | CIFAR-10 | CelebA |
> |-----------------|----------|--------|
> | **DDPM-train**  | 278      | 435    |
> | **DDSM-train**  | 502      | 1036   |
> | **DDSM-search** | **320**  | **524**|
>
> **Table:** Total GPU hours of DDSM.
>
> ---
>
> [A] OMS-DPM: Optimizing the Model Schedule for Diffusion Probabilistic Models
>
> [B] AutoDiffusion: Training-Free Optimization of Time Steps and Architectures for Automated Diffusion Model Acceleration
>
> [C] Denoising Diffusion Step-aware Models

---

### Official Review · Reviewer_wo3A · 2024-07-09

**Soundness:** 3
**Presentation:** 2
**Contribution:** 2
**Rating:** 5
**Confidence:** 5

**Summary:**

This paper presents an approach to accelerate diffusion model inference by capitalizing on the minimal change in encoder features across time steps. The proposed encoder propagation strategy reduces encoder computations by reusing encoder features from previous time steps. The proposed method is comparable to existing acceleration methods and demonstrates effective acceleration across diverse tasks.

**Strengths:**

1.	The proposed method offers a promising solution to speed up diffusion models.

2.	The experiments are conducted from various tasks, e.g., text to image; text to video; personalized generation and reference-guided generation.

**Weaknesses:**

1.	The evaluation metrics are insufficient, as concerns exist regarding the universality of FID in assessing generative models. It is recommended to validate performance on additional metrics, such as the newly proposed ImageReward [1] and Pick Score [2].

2.	It is unclear whether the latency is comparable when fewer sampling steps are used (Table 11 and Table 12). Moreover, the performance improvement is marginal compared to results with fewer sampling steps, and mainly stems from prior noise injection, which is not the core contribution of the paper. I am also curious whether fewer sampling steps, if equipped with prior noise injection, could achieve similar performance as the results presented in this paper.

3.	Comparing multi-GPU results with those single-GPU results from DeepCache seems unfair (Table 1). Besides, the acceleration outcomes on a single GPU are marginally improved when compared with other acceleration methods.

4.	The DiT does not inherently include concepts of Encoder and Decoder; thus, categorizing based solely on observations into Encoder and Decoder is somewhat imprecise.

5.	The layout of the paper is somewhat disorganized; the order of tables does not align with the sequence of their introduction in the text (e.g., Table 1 and Table 3), making it hard to follow.

[1] Xu J, Liu X, Wu Y, et al. Imagereward: Learning and evaluating human preferences for text-to-image generation. NeurIPS 2023.

[2] Kirstain Y, Polyak A, Singer U, et al. Pick-a-pic: An open dataset of user preferences for text-to-image generation. NeurIPS 2023.

**Questions:**

See weakness for details.

**Limitations:**

The authors have discussed both limitations and potential negative social impacts.

---

> ### Author Rebuttal · Authors · 2024-08-07
>
> We appreciate your feedback and will incorporate the discussions mentioned below to enhance the quality of our paper. Note that we utilize the numerical references to cite sources within the main paper.
>
> **W1. Additional metrics**
>
> Thanks for your advice. We showcase our experimental results with the ImageReward and PickScore evaluation metrics over the MS-COCO2017 10K dataset, as shown in the **Table below**.
> Our method FasterDiffusion enhances sampling efficiency while preserving the original model performance.
>
> |                           | **ImageReward**$\uparrow$ | **PickScore**$\uparrow$ | **s/image** |
> |----------------------------|---------------------------|-------------------------|-------------|
> | **SD (DDIM)**              | 0.149                     | 52.10%                  | 2.42        |
> | **SD (DDIM) w/ Ours**      | 0.162                     | 47.90%                  | 1.42        |
>
> **Table:** Metrics in ImageReward and PickScore.
>
> ---
>
> **W2. Few step generation with noise injections**
>
> To extend the content illustrated in Table 11 and Table 12,
> we present the full results with fewer sampling steps in **Table below**, specifically with 9 and 25 steps as in Table 11.
>
> We mainly present the generation results with 9 steps because it is the number of $key$ time-steps for the 50-step SD model determined in our approach.
> We increase the number of sampling steps (i.e., 25 steps),  and find that the sampling results do not perform as well as DDIM with 50 steps and its variation with FasterDiffusion (Figure.12).
> This demonstrates that our method is not simply reducing the number of sampling steps.
>
> As shown on **Table below**,  we conduct
> DDIM scheduler inference time with various steps.
> Although FasterDiffusion is slightly longer than DDIM scheduler with 9 steps, our sampling results are much closer the 50-step DDIM generation quality, whereas the 9-step DDIM results are much inferior (see Table.11, Table.12, and Figure.12 for examples).
>
> For directly applying noise injection to the generation phase, we show image generation examples in **Figure.27** in the rebuttal file.
> Fewer sampling steps equipped with prior noise injection result in almost no improvement in image quality. **Table below** further supports this conclusion.
> While in our case, FasterDiffusion working without the noise injection will lead to smoother textures, as shown in Figure.6.
>
> The noise injection technique helps in preserving fidelity in the generated results.
>
> | Sampling method   | T  | **FID** $\downarrow$ | **Clipscore** $\uparrow$ | **s/image** $\downarrow$ |
> |------------------------------|----|-----------------------|--------------------------|--------------------------|
> | **DDIM**                     | 50 | 21.75                 | 0.773                    | 2.42              |
> | **DDIM**                     | 25 | 22.16                 | 0.761                    | 1.54                     |
> | **DDIM w/ noise injection**  | 25 | 21.89                 | 0.761                    | 1.54                     |
> | **DDIM**                     | 9  | 27.58                 | 0.735                    | 0.96                     |
> | **DDIM w/ noise injection**  | 9  | 27.63                 | 0.736                    | 0.96                     |
> | **DDIM w/ ours**             | 50 | **21.08**                 | **0.783**                    | **1.42**               |
>
> **Table:** Quantitative comparison for DDIM with fewer steps.
>
> ---
>
> **W3. Comparison with single-GPU methods**
>
> Our advantage over other methods (e.g., DeepCache) is the ability to perform inference on non-key time steps in parallel, therefore enabling parallel processing across multi-GPUs.
> This characteristic assist our method FasterDiffusion to achieve better speedup performance compared to other methods.
>
> In this paper, we notice that DeepCache is not parallelizable on multiple GPUs since DeepCache needs to use all or part of the encoder and decoder at every time step.
> FasterDiffusion, on the other hand, only uses the encoder at the $\textit{key}$ time steps, which enables parallel processing at non-key time steps and faster inference time cost.
> It is also worth to notice that FasterDiffusion can be further combined with a wide range of diffusion model based tasks, including Text2Video-zero, VideoFusion, Dreambooth, and ControlNet. In this case, DeepCache often shows much slower speed while applied to these tasks. As an example shown in **Table.14 (in 'global' response)**, DeepCache applied to the ControlNet is slower by 24\% compared to the FasterDiffusion based ControlNet.
>
> More specifically, since the ControlNet model requires an additional encoder, our method FasterDiffusion is able to execute this extra encoder in a parallel manner and reuse it, and that makes the additional time negligible.
> On the other hand, DeepCache is reusing the decoder feature, which leads the ControlNet to wait for the additional encoder to complete computation at each time-step, resulting in almost no time saved from skipping the encoder in the standard SD models.
>
> Please refer to our 'global' response (**General Response 1**) for **more details**.
>
> **W4. Applicability to DiT based models**
>
> The DiT includes 28 transformer blocks.
> Through visualization and statistical analysis (see **Figure.30** in the rebuttal PDF and **Figure.11** in main paper),
> we observe that the features in the first several transformer blocks change minimally, similar to the Encoder in SD, while the features in the remaining transformer blocks exhibit substantial variations, akin to the Decoder in SD.
> For ease of presentation, we refer to the first several transformer blocks of DiT as the Encoder and the remaining transformer blocks as the Decoder. In **Table 2**, we demonstrate the accelerating performance of our method while applied to DiT-based generation models.
>
> **W5. Paper layout**
>
> We will carefully design the layout in the future version for better reading experience.

---

> > ### Comment · Reviewer_wo3A · 2024-08-13
> >
> > Thank you for your response.
> > I'm still a bit confused, as typically, the PickScore should be a specific numerical value, not a percentage.

---

> > > ### Author Response · Authors · 2024-08-13
> > >
> > > Dear Reviewer wo3A:
> > >
> > > We are grateful for your expeditious response and we are glad to clarify and address your concerns further.
> > >
> > > For the PickScore, we used the calculation method recommended by the official github repository reference:
> > >
> > > > For a prompt $x$ and a image $y$，the scoring function $s$ computes a real number by representing $x$ using a transformer text encoder and $y$ using a transformer image encoder as d-dimensional vectors, and returning their inner product:
> > > $$
> > > s(x,y) = E_{txt}(x) · E_{img}(y) · T
> > > $$
> > > Where $T$ is the learned scalar temperature parameter of CLIP.
> > >
> > > Following that, assuming the original image generated by SD is $y_1$, and the image generated after combining with Ours is $y_2$, we can calculate the preference distribution vector $p$:
> > > $$
> > > p_i = \frac{\exp  s(x, y_i)}{\sum_{j=1}^2 \exp s(x,y_j)}
> > > $$
> > > PickScore official recommended get probability, if there are multiple images available for selection.
> > >
> > > We also calculated the original PickScore scores, as shown in the table below:
> > >
> > > |                  | PickScore$\uparrow$ | s/image |
> > > | ---------------- | ------------------- | ------- |
> > > | SD(DDIM)         | 21.43               | 2.42    |
> > > | SD(DDIM) w/ Ours | 21.35               | 1.42    |
> > >
> > > Our method FasterDiffusion enhances sampling efficiency while preserving the original model performance.
> > >
> > > We hope that these clarifications have addressed your concerns satisfactorily and look forward to any further feedback you may have. We are committed to enhancing our work based on your expert guidance.

---

> > > > ### Comment · Reviewer_wo3A · 2024-08-13
> > > >
> > > > Thank you to the authors for the detailed response. Most of my concerns have been addressed, so I have raised my score.

---

### Official Review · Reviewer_BYFv · 2024-07-12

**Soundness:** 3
**Presentation:** 2
**Contribution:** 2
**Rating:** 5
**Confidence:** 4

**Summary:**

This paper presents an extensive study of the evolution of internal activations in diffusion U-Nets and uses their findings to motivate a training-free approach for accelerating sampling from diffusion models. The method is demonstrated to successfully speed up inference in a variety of settings, including different architectures (including non-hierarchical DiTs), different samplers, and some modified inference processes.

**Strengths:**

- The method is demonstrated to work well on a wide range of networks & tasks, including U-Net-based LDMs (SD), Imagen-style cascaded pixel-space U-Net diffusion models (IF), and DiTs. Especially DiT is notable, as previous methods do not work on homogeneous transformer architectures out-of-the-box.
- The extensive appendix provides a lot of useful additional information and experimental results and supplements the empirical study about the evolution of internal features in diffusion models well.
- The proposed method opens up avenues for further accelerating sampling speed as measured in wall clock delay for single samples by utilizing multiple GPUs to distribute the proposed parallel encoder propagation strategy.
- The proposed "prior noise injection method" helps alleviate artifacts incurred from the optimized sampling process.

**Weaknesses:**

- No details are provided about the user study, and false claims relating to it are made in the checklist [questions 14 and 15].
- The main method seems to only be a minor incremental evolution of DeepCache's methodology (encoder features are cached and reused without adaptation of features or network in following steps, optionally multiple times, with the main difference being what is defined as the encoder), and seems to provide a *reduced* speedup when compared in a fair single-GPU setting [Tab. 1, Sec. 4.1 l. 271: 41% speedup vs. 56% speedup, at comparable quality levels].
- The presentation of the paper needs some work. Font sizes in some figures are excessively small [cf. Figs. 1, 2, 3, 7], Tabs. 2 and 3 are excessively compressed up to a point at which readability is impaired. The paper would also benefit from a thorough re-read and corrections for the camera-ready version. Especially grammar errors that substantially affect the claims made in the paper (e.g., sampling time reduction "to 41%" [l. 271] instead of "by 41%") and typos that change the meaning of words (e.g., "rudely" [l.105]), but also names of methods (e.g., "DeepFolyd-IF" [l. 60], "Stable Diffuison" [l. 525]).

**Questions:**

- Can the authors provide examples and evidence of practically relevant settings where DeepCache fails or underperforms significantly compared to the proposed method (such as a combination with DPM-Solver(++), ControlNet, or Dreambooth etc)? These would help address the main method-related weakness mentioned.
- What do the ellipses in the key time step steps mean? Is every time step between 80 and 25 included for $\{80,...,25\}$?
- Is there a relevant difference between the proposed "prior noise injection method" and the various methods that artificially increase the noise level slightly during ODE sampling, such as ``churn'' in (Karras et al., Elucidating the Design Space of Diffusion-Based Generative Models, NeurIPS 2022)?
- Are all the videofusion samples in the supplementary material just corrupted or is there something to be demonstrated there?
- Does parallel-batch encoder propagation limit the possible batch size on a single GPU compared to other sampling methods? If so, could the authors provide a performance comparison that individually maxes out the batch size for the various approaches (including competing ones) and then reports the samples/second? As generating multiple variants simultaneously is a standard approach when generating images with diffusion models in practice, this is an important avenue for optimization. Specifically, I am worried that at a maximal batch size, parallel encoder propagation will only offer a negligible speedup compared to encoder propagation (due to larger generation batch sizes being possible with non-parallel encoder propagation, as VRAM usage is substantially different between the two methods [Tab. 8]), at which point this method would provide less than half of the speedup provided by prior art.

**Limitations:**

The authors have adequately addressed the limitations of their work.

---

> ### Author Rebuttal · Authors · 2024-08-07
>
> We appreciate your feedback and will incorporate the discussions mentioned below to enhance the quality of our paper. Note that we utilize the numerical references to cite sources within the main paper.
>
> **W1. User study details.**
>
> The study participants were volunteers from our college. The questionnaire consisted of 35 questions, each presenting two images: one from the baseline methods (including ControlNet, SD, DeepFloyd-IF, DPM-Solver++, Dreambooth, Custom Diffusion, Text-to-Video, etc.) and the other from our method FasterDiffusion (one example shown in **Figure.29**).
>
> Users were required to select the image where the target was more accurately portrayed, or choose “both equally good”. A total of 18 users participated the questionnaire, resulting in totally 630 samples (35 questions × 1 option × 18 users). As the final results shown in **Figure.8**, the chosen percentages for SD, DeepFloyd-IF, DPM-Solver++, Text-to-Video, ControlNet, and Personalize were 48\%, 44\%, 51\%, 52\%, 46\%, and 39\%, respectively.
> These results show that our method  performs on par with the baseline methods.
>
> **W2\&Q1. Comparison with DeepCache**
>
> DeepCache is developed based on the observation over the temporal consistency between high-level features.
> In this paper, we have a more thorough analytical study over the SD model features as shown in **Figure.3**, where we find out the encoder features change less, whereas the decoder features exhibit substantial variations across different time steps. This insight motivates us to omit encoder computation at certain adjacent time-steps and reuse encoder features of previous time-steps as input to the decoder in multiple time-steps. We further determine the key time steps in the T2I inference stage, which helps our method to skip the time steps in a more scientific way.
>
> It is also important to note that DeepCache is not parallelizable on multiple GPUs since DeepCache needs to use all or part of the encoder and decoder at every time step.
> FasterDiffusion, on the other hand, only uses the encoder at the $\textit{key}$ time steps, which enables parallel processing at $\textit{non-key}$ time steps and faster inference time cost.
>
> It is also worth to notice that FasterDiffusion can be further combined with a wide range of diffusion model based tasks, including Text2Video-zero, Dreambooth, and ControlNet. In this case, DeepCache often shows much slower speed while applied to these tasks. As an example shown in **Table below**, when combined with  ControlNet, DeepCache  is slower by **24\%** compared to the FasterDiffusion.
>
> More specifically, since the ControlNet requires an additional encoder, our method FasterDiffusion is able to execute this extra encoder in a parallel manner and reuse it, and that makes the additional time negligible.
> On the other hand, DeepCache is reusing the decoder feature, which leads the ControlNet to wait for the additional encoder to complete computation at each time-step, resulting in almost no time saved from skipping the encoder in the standard SD models.
> As shown in **Figure.28**, FasterDiffusion successfully preserves the given structure information and achieves similar results as the original ControlNet.
>
> |          | **Clipscore**$\uparrow$ | **FID**$\downarrow$ | **s/image**$\downarrow$ |
> |--------------------------|-------------------------|---------------------|-------------------------|
> | **ControlNet**           | 0.769                   | 13.78               | 3.20                    |
> | **ControlNet w/ DeepCache** | 0.765                   | 14.18               | 1.89 (1.69x)            |
> | **ControlNet w/ Ours**   | 0.767                   | 14.65               | **1.52 (2.10x)**        |
>
> **Table 14:** When combined with ControlNet (Edge) 50-step DDIM, our inference time shows a significant advantage compared to DeepCache.
>
> ---
>
> **W3. Paper presentation**
>
> We will carefully correct these figures, tables, grammar errors and typos in the future version.
>
> **Q2. Ellipsis meaning in the key time step**
>
> The ellipsis in $t^{key}$ denotes each time-step between the time-steps on either side of the ellipsis. For example, 80...25 means that every time-step between 80 and 25 is included.
>
> **Q3. Relevant difference**
>
> In EDM [A] (Karras et al.), they increase the noise level during ODE sampling to improve deterministic sampling, which is referred to as stochastic sampling (i.e., 'churn'). Compared to deterministic sampling, it injects noise into the image at each step, which helps to enhance the quality of the generated images. Song et al. [B] first observed that perturbing data with random noise makes the data distribution more amenable to score-based generative modeling.  Increasing the noise level in EDM is based on Song's paper.
>
> Their purpose for injecting noise is to perturb the data, whereas our purpose for injecting noise during sampling is to preserve high-frequency details in the image during the denoising process, preventing the diffusion model from removing high-frequency information as noise (see **Figure.6** in the main paper).
> In addition, our method FasterDiffusion differs from them by only inserting noise in the later stage of the diffusion steps, while they insert noise to all time-steps.
>
> **Q4. Video corruption**
>
> The files (including the videofusion samples) in the supplementary material are not corrupted. They can be opened correctly using the default video players on Windows or Mac operating systems.
> However, on the Linux operating systems with graphical interfaces (e.g., Ubuntu), the default video player cannot open these files normally. You may need to download an alternative video player from the software store (e.g., Celluloid) to read these videos.
>
> **Q5. Parallel or Serial**
>
> Please refer to our 'global' response (**General Response 1**) for details.
>
> ---
>
> [A] Elucidating the design space of diffusion-based generative models
>
> [B] Generative modeling by estimating gradients of the data distribution

---

> > ### Comment · Reviewer_BYFv · 2024-08-09
> >
> > > **W1. User study details.**
> > >
> > > The study participants were volunteers from our college. The questionnaire consisted of 35 questions, each presenting two images: one from the baseline methods (including ControlNet, SD, DeepFloyd-IF, DPM-Solver++, Dreambooth, Custom Diffusion, Text-to-Video, etc.) and the other from our method FasterDiffusion (one example shown in Figure.29).
> > >
> > > Users were required to select the image where the target was more accurately portrayed, or choose “both equally good”. A total of 18 users participated the questionnaire, resulting in totally 630 samples (35 questions × 1 option × 18 users). As the final results shown in Figure.8, the chosen percentages for SD, DeepFloyd-IF, DPM-Solver++, Text-to-Video, ControlNet, and Personalize were 48%, 44%, 51%, 52%, 46%, and 39%, respectively. These results show that our method performs on par with the baseline methods.
> >
> > Thanks for providing these details and the screenshots in the rebuttal PDF, but this does still not resolve the major concerns about question 15 (IRB board approval, potential risks for participants).

---

> > > ### Author Response · Authors · 2024-08-09
> > > **IRB considerations**
> > >
> > > I appreciate your emphasis on the significance of IRB approvals for user studies in current computer vision research. It is indeed crucial to ensure that research involving human subjects adheres to ethical standards and guidelines, and obtaining IRB approval is a key step in this process.
> > >
> > > The Institutional Review Board (IRB) is a formalized entity tasked with upholding research ethics by conducting a rigorous assessment of the methodologies proposed for research endeavors that involve human subjects. The primary objective of these reviews is to safeguard the wellbeing of study participants, ensuring that they are not subjected to harm and that their rights and welfare are duly protected. This is achieved through a meticulous examination of research protocols and associated documentation, with the intention of identifying and mitigating potential risks.
> > >
> > > In the context of this research paper, it is noteworthy that the study design does not pose any risks of harm to the participants. Instead, we employ a user study methodology wherein participants are invited to express their opinions regarding the images generated by our approach in comparison to those produced by alternative methods. To further safeguard the participants, we have implemented rigorous safety checks within our image generation pipelines, ensuring that no harmful or inappropriate content is included in the output images. Consequently, the study adheres to ethical principles by minimizing potential risks and prioritizing the protection of participant rights and welfare.
> > >
> > > In this context, we address checklist item 15 as outlined by the regulatory body, as our study does not involve research concerning human subjects. Rather, it may be construed as an online questionnaire administered for data collection purposes.
> > >
> > > We endeavored to address all your concerns comprehensively to ensure clarity in the explanations provided.  If you have further questions about our research, please feel free to engage with us for further discussion.

---

> > > > ### Comment · Reviewer_BYFv · 2024-08-12
> > > >
> > > > Given the authors' responses to my concerns about the user study, my main concerns relating to it and the missing material in the paper have been resolved. To the best of my knowledge, the user study performed by the authors qualifies as "research involving human subjects" in most jurisdictions, but this might not be the case in theirs, and I am no ethics reviewer who would have the qualification to judge in a more precise manner. I would still urge the authors to double-check that they followed all the rules set by their institutions, country, and NeurIPS, and expect them to include all relevant details, especially those presented as a part of the rebuttal, in the final version.

---

> > > > > ### Comment · Reviewer_BYFv · 2024-08-12
> > > > >
> > > > > After carefully reviewing the authors' rebuttal and review responses, only some of my concerns have been resolved.
> > > > >
> > > > > One important remaining concern is comparisons in fair settings.
> > > > > The authors were able to address my concern about a reduced single-GPU speed compared to the previous DeepCache in the case of using ControlNets, but not in general. More importantly, as far as I could see, the authors still did not provide any results that max out the batch size per GPU, as suggested in my initial review. I think this is an essential comparison, as VRAM usage is one of the main practical limitations for practical deployments, both due to the cost of GPUs and, more importantly, because larger batch sizes typically make inference more efficient. This is relevant even if only single images are generated per user, as any standard inference backend will batch multiple calls for efficiency. The fact that the method proposed in this work shows a substantial VRAM increase would suggest that this method's speed is likely limited if one increases the batch size to the maximum possible on a reasonably sized GPU. Additionally, I disagree that the authors' argument regarding multi-GPU inference is practically relevant. Many practical inference systems generate multiple images at once (at which point one can just split up the samples over multiple GPUs instead of splitting up the inference process of each sample over multiple GPUs). In other cases, [unless one is already using a top-of-the-line available GPU, which is heavily non-standard for diffusion production loads], one can likely increase the inference speed by a similar amount at the same cost by just using one higher-end GPU instead of two GPUs.
> > > > >
> > > > > I was now also able to open the problematic Videofusion example videos from the supplementary material on a Windows machine using the standard media player. My issues with opening them persist on the two different MacOS-based systems I tried opening them with initially (using the standard viewer as suggested by the authors). If the authors are aware of any uncommon codec choices etc that might have affected this, I'd recommend that they switch to a standard encoding for future submissions to prevent these problems. Most importantly, I have been able to look at the samples now, and they seem reasonable.
> > > > >
> > > > > As the authors have addressed my major critical concerns about the user study and some minor concerns but left my major concerns about fair comparisons insufficiently addressed, I'm raising my score from 3 to 5. With aforementioned shortcomings, I think the main value this work provides in its current form is the analysis of the network, but I'm sceptical about the practical relevance of the proposed method to exploit them in general use cases.

---

> > > > > > ### Author Response · Authors · 2024-08-13
> > > > > >
> > > > > > We are grateful for your prompt response, and we are glad to clarify and address your concerns further.
> > > > > >
> > > > > > **User study details and its IRB concerns**
> > > > > >
> > > > > > Thanks for your attention to the user study, particularly in relation to the IRB rules. In any future version, we are committed to incorporating detailed experimental information on the user study accompanied by a thorough explanation of IRB approvals, which we already addressed in the rebuttal phase.
> > > > > >
> > > > > > **Single-GPU inference time comparison**
> > > > > >
> > > > > > To further relieve your concerns over the single-GPU setting, we conducted experiments under the same setup. Our observation indicates that SD1.5 model attains a peak batch size of 30, with an average inference time of 1.903 seconds per image.
> > > > > > Upon integration with our approach FasterDiffusion, the maximum batch size experiences a minor reduction to 25, but the inference time per image demonstrates a significant decrease to 1.551 seconds.
> > > > > > These results highlight that, despite the constraints of the maximum batch size per GPU, our method delivers a notable enhancement in inference speed. FasterDiffusion presents a robust solution for navigating time-space trade-offs effectively.
> > > > > >
> > > > > > **Video format issue**
> > > > > >
> > > > > > We appreciate your valuable feedback and the efforts you made in examining our video examples.  We will investigate this matter thoroughly and guarantee the application of standard encoding in any future submissions.
> > > > > > We are glad to hear that you were able to review the samples and found them reasonable.
> > > > > > In the future version, we aim to release a project page where all examples, including the Videofusion samples, are easily accessed for convenience.

---

> > > > > > > ### Comment · Reviewer_BYFv · 2024-08-13
> > > > > > >
> > > > > > > Thank you for providing the additional evaluation for the max batch size setting. In my opinion, the presented speedup of ~20% (down from the 41% presented in the paper, which was already lower than that of other methods) in this fair evaluation setting is not particularly favorable for the method proposed in this work. Thus, I will retain my initial increased score.

---

### Official Review · Reviewer_wXBv · 2024-07-12

**Soundness:** 3
**Presentation:** 3
**Contribution:** 3
**Rating:** 6
**Confidence:** 4

**Summary:**

This method can be applied at inference time without requiring retraining to improve sampling efficiency while maintaining high image quality and can be combined with other approaches to speed up diffusion models. The approach is effective across various conditional diffusion tasks, including text-to-video generation (e.g., Text2Video-zero, VideoFusion), personalized image generation (e.g., Dreambooth), and reference-guided image generation (e.g., ControlNet). Using encoder features from previous time steps as input for decoding multiple later time steps allows for concurrent decoding, further accelerating SD sampling by 41%, DeepFloyd-IF sampling by 24%, and DiT sampling by 34%. Besides, a prior noise injection strategy is introduced to preserve image quality, maintaining texture details in the generated images.

**Strengths:**

Novelty: The three techniques to design efficient diffusion models are novel.

Significance: The problem of efficient diffusion transformer inference acceleration is significant in diffusion models.

Methodology: The proposed algorithm is well-formulated and well-explained.

Results: The experimental results show significant improvements over existing methods over SD, DiT, and DeepFloyd-IF and are widely applied to text-to-image, text-to-video generation, and personalization.

**Weaknesses:**

The visualization results for smaller steps appear suboptimal, potentially due to the absence of consistency models or adversarial distillation models for verification. This raises concerns about the foundational assumption of maintaining high similarity throughout the process. It is crucial to explore whether incorporating consistency models or adversarial distillation methods could enhance the quality of results and ensure the assumption of high similarity holds even in smaller steps. Providing more robust verification methods would strengthen the validity and reliability of the visual outputs generated by the proposed approach.

**Questions:**

Extra Cost to Identify Key Time-Steps: What are the additional costs associated with identifying key time steps in the diffusion process? Is this identification process capable of being learned dynamically, or is it independent of the data? Understanding the computational overhead or resource allocation needed for this identification process is crucial for evaluating the overall practicality of the method.

Enhancing Application Area with Smaller Steps Diffusion Models: Applying this approach to smaller steps diffusion models has the potential to significantly expand its applicability. By effectively handling smaller time steps, the method could be utilized in a wider range of scenarios, enhancing its versatility and impact.

**Limitations:**

Please refer to the weakness and questions part.

---

> ### Author Rebuttal · Authors · 2024-08-07
>
> We appreciate your feedback and will incorporate the discussions mentioned below to enhance the quality of our paper. Note that we utilize the numerical references to cite sources within the main paper.
>
> **W1\&Q2: Cooperation with few-step T2I models**
>
> In the **Limitations** part of the main paper, we present that our method faces challenges in generating quality results when using a smaller number of sampling steps.
> The underlying reason is that our method FasterDiffusion is accelerating the T2I generative model based on the empirical study over the feature changes at adjacent time-steps in multistep DMs (e.g., Stable Diffusion with 50 steps), as shown in Fig.3. We can observe that the encoder features change minimally, whereas the decoder features exhibit substantial variations across different time steps. This insight motivates us to omit encoder computation at certain adjacent time-steps and reuse encoder features of previous time-steps as input to the decoder in multiple time-steps.
>
> To check the possibility to combine our method with few-step T2I models, including LCM [A] and SDXL-Torbo [B], we visualize the features for both the encoder and decode of UNet of these models.
> In 4-step LCM, the features of the encoder and decoder of UNet vary greatly at adjacent time steps (see **Figure.22**). The feature changes of the encoder are not as subtly varied as they are in 50-step Stable Diffusion models, which means that the encoder features cannot be shared in adjacent time-steps  (see **Figure.24**).
>
> The 4-step SDXL-Turbo [A] shares the same problem as the 4-step LCM, as can be seen in **Figure.23** and  **Figure.25**.
> Therefore, these few-step T2I generative models (LCM and SDXL-Turbo) cannot be combined with FasterDiffusion
>
> However, as a future task, we aim to apply these findings to model distillation. We hypothesize that incorporating  the encoder only once and the decoder across multiple time-steps can help capture richer semantic information and accelerate the diffusion model distillation training.
> For example, in **Figure.26**, we illustrate the comparison between 1-step LCM and 4-step LCM feature visualization, the decoder feature maps are with more diversity with 4 steps. This implies that we can share one encoder by four decoders to capture the semantic information encapsulated in these decoders. By this means, we can achieve faster distillation for the T2I models and quicker parallel T2I generations.
> In this paper, we focus only on training-free acceleration techniques and will make this training-required distillation method our future work.
>
> **Q1: Additional cost to identify key time steps**
>
> To identify key time steps, we conducted empirical analysis of feature changes at adjacent time steps in multistep Stable Diffusion models (50-step model as an example).
> This analysis is based on statistics from the distribution of features across 100 random prompts, which does not impose a high time cost.
>
> Based on analysis as shown in **Section 3.2**, we determine the $\textit{key}$ time-step as $t^{key}=\\{50, 49, 48, 47, 45, 40, 35, 25, 15\\}$ for the 50-step Stable Diffusion model.
> As a general case, we also applied this $\textit{key}$ time-step configuration to the ControlNet [10], Text2Video-zero [4], VideoFusion [5], Dreambooth [7] and Custom Diffusion [8] models based on the 50-step SD models.
>
> Moreover, once we determine the $\textit{key}$ time steps through analysis, the T2I diffusion models and their applications using FasterDiffusion always employ the same determined $\textit{key}$ time steps.
> This is a **once-for-all** solution, with no additional costs for each application scenario.
>
> ---
>
> [A] Latent consistency models: Synthesizing high-resolution images with few-step inference.
>
> [B] Adversarial diffusion distillation.

---

### Author Rebuttal · Authors · 2024-08-07

**“global” response**

We appreciate all reviewers (**R1=wXBv**, **R2=BYFv**, **R3=wo3A**, **R4=i4aK**) for their positive feedbacks. They note that this paper is well-written (R4) and easy to understand (R4); the technique is novel (R1) and promising (R3); that the proposed algorithm is well-formulated and well-explained (R1); that we present insightful analysis of UNet (R4); that we provide extensive experiments over various network architectures (R1, R2, R4), tasks (R1, R2, R3), and use various metrics (R4); that we can also work on diffusion transformer inference acceleration (R1,R2). Below we respond to general questions raised by reviewers; We use **W** to abbreviate **Weaknesses**, **Q** to represent **Questions** and **L** for **Limitations**.
 Note that we utilize the numerical references to cite sources within the main paper.

**General Response 1. Parallel on multi-GPU or Serial on single-GPU (R2-Q5, R3-W3)**

Our advantage over other methods (e.g., DeepCache) is the ability to perform inference on *non-key* time steps in parallel, therefore enabling parallel processing across multi-GPUs.
This characteristic assist our method FasterDiffusion to achieve better speedup performance compared to other methods.
It is also noticeable that DeepCache is not parallelizable, since it needs to use all or part of the encoder and decoder at every time step. Due to this reason, DeepCache cannot be further improved by deploying on multiple GPUs.

Deep learning models rely heavily on parallel training and inference across multi-GPUs. Using only a single GPU can result in excessive time consumption while dealing with large models. Therefore， it is a very relevant and impactful contribution that parallelizing existing algorithms and unlocking the speed-up available due to multiple GPU computations. In this paper, our proposed FasterDiffusion parallelizes the processing of multiple time-steps, enhancing processing efficiency.


In addition, our findings can also be applied to model distillation by using the decoder at multiple time steps to capture richer semantic information, which DeepCache cannot achieve. As a future task, we aim to apply these findings to model distillation. We hypothesize that incorporating the encoder only once and the decoder across multiple time-steps can help capture richer semantic information, and accelerate the diffusion model distillation training.
For example, in **Figure.26** in rebuttal PDF, we illustrate the comparison between 1-step LCM and 4-step LCM feature visualization, the decoder feature maps are with more diversity with 4 steps.
This implies that we can share one encoder by 4-step decoder to capture the semantic information encapsulated in 4-step decoder. By this means, we can achieve faster distillation for the T2I models and quicker parallel T2I generations.
In this paper, we focus only on training-free acceleration techniques and will make this training-required distillation method our future work.

|          | **Clipscore**$\uparrow$ | **FID**$\downarrow$ | **s/image**$\downarrow$ |
|--------------------------|-------------------------|---------------------|-------------------------|
| **ControlNet**           | 0.769                   | 13.78               | 3.20                    |
| **ControlNet w/ DeepCache** | 0.765                   | 14.18               | 1.89 (1.69x)            |
| **ControlNet w/ Ours**   | 0.767                   | 14.65               | **1.52 (2.10x)**        |

**Table 14:** When combined with ControlNet (Edge) 50-step DDIM, our inference time shows a significant advantage compared to DeepCache.

---

### Decision · Program_Chairs · 2024-09-25

**Decision:**

Accept (poster)

**Comment:**

The manuscript has been reviewed by four reviewers, all of whom, after the rebuttal, gave positive ratings (including borderline positive).

In general, the reviewers are happy with the novelty, motivation, significance, and experimental validation.

The AC agrees with the consensus, and recommends the manuscript to be accepted. Congrats!